# MathEval: A Comprehensive Benchmark for Evaluating Large Language Models on Mathematical Reasoning Capabilities

## Abstract

Mathematical reasoning is a fundamental aspect of intelligence, encompassing a spectrum from basic arithmetic to intricate problem-solving. Recent investigations into the mathematical abilities of large language models (LLMs) have yielded inconsistent and incomplete assessments. In response, we introduce MathEval. A comprehensive benchmark designed to methodically evaluate the mathematical problem-solving proficiency of LLMs across varied contexts, adaptation strategies, and evaluation metrics. MathEval consolidates 22 distinct datasets, encompassing a broad spectrum of mathematical disciplines, languages (including English and Chinese), and problem categories (ranging from arithmetic and competitive mathematics to higher mathematics), with varying degrees of difficulty from elementary to advanced. In order to surmount the complexities associated with mathematical reasoning output, which lacks a unified pattern for discerning the true answers, and to adapt to the outputs of various models and prompts, we propose the utilization of GPT-4 as an automated pipeline for answer extraction and comparison. To broaden the utility of MathEval beyond the scope of GPT-4, we have harnessed the extensive results from GPT-4 to train a deepseek-7B-based answer comparison model, enabling precise answer validation for those without access to GPT-4. This model will also be made publicly available. To mitigate potential test data contamination and truly gauge progress, MathEval incorporates an annually refreshed set of problems from the latest Chinese National College Entrance Examination (Gaokao 2023, Gaokao 2024), thereby benchmarking genuine advancements in mathematical problem solving skills.

## 1 Introduction

Mathematics stands as a cornerstone to human intelligence (Ahn et al., 2024), encompassing a comprehensive range of abilities from basic arithmetic to reasoning. Recently, there has been a rapid increase in research related to large-scale language models (LLMs) in mathematics, which has swiftly propelled the enhancement of LLMs' mathematical reasoning abilities. However, the evaluation of these models remains challenging due to three primary issues: **"incomprehensiveness"**, **"inadequate adaptation"** to varying model types and datasets, and **"inconsistency"**.

**"Incomprehensiveness"** indicates that evaluations often do not cover a wide array of datasets, neglecting factors such as language diversity, problem types, and complexity levels. This limited scope can skew perceptions of a model's versatility and effectiveness. **"Inadequate adaptation"** highlights the shortcomings in current evaluations to flexibly accommodate different types of models and datasets. For instance, chat models, which have been fine-tuned during the alignment phase, are especially sensitive to the structure of prompts. Similarly, evaluations should also adapt prompts to fit the specific characteristics of each dataset.

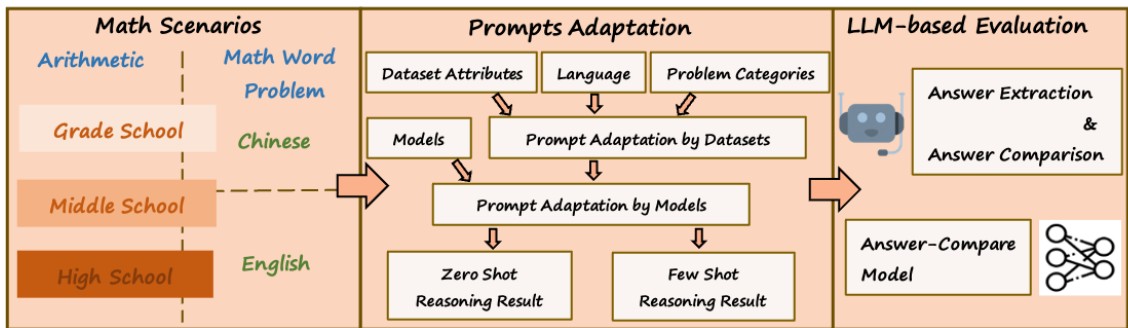

Figure 1: Three core components of MathEval addressing key challenges. MathEval integrates: (1) Math Scenarios, which encompass languages (Chinese and English), problem types (arithmetic and math word problems), and educational levels (primary, middle, and high school) to comprehensively address the challenge of incomprehensiveness; (2) Prompt Adaptation, which selects tailored dataset and model templates based on specific dataset characteristics and model information, effectively tackling the problem of inadequate adaptation; (3) LLM-based Evaluation, utilizing GPT-4 for answer extraction and comparison to mitigate inconsistency issues, with an alternative distilled compare-answer model available for users without access to GPT-4. This structure ensures a robust and fair evaluation of LLMs' mathematical reasoning capabilities.

For example, multiple-choice problems may require prompts that include hints to guide the selection from provided options, whereas math word problems might benefit from prompts that encourage Chain-of-Thought (CoT) reasoning. **"Inconsistency"** arises when the same model yields different performances on identical datasets, complicating the accurate estimation of its true capabilities. This issue primarily stems from the difficulty in verifying answers to mathematical word problems, where outputs may include reasoning steps, equations, and final answers in various formats (e.g., 1/2 and (\frac{1}{2})). Extending this to different models and various types of datasets further complicates the evaluation. Rule-based methods for extracting and comparing answers, commonly utilized in benchmarks such as OpenCompass (Contributors, 2023) and HELM (Liang et al., 2023), often lack robustness. Even minor modifications can significantly alter the evaluation outcomes, making it impractical to tailor these rules for each specific model and dataset. Consequently, standardizing the process of extracting and comparing outputs continues to pose a significant challenge in benchmark evaluations. More related works are discussed in Section 4.

To address these challenges, we introduce **MathEval**, a comprehensive and unified benchmarking framework, as illustrated in Figure 1. MathEval incorporates 22 datasets in both Chinese and English, covering a wide range of mathematical problems from primary to high school levels, and includes a dynamically updated dataset to prevent test data contamination. Each dataset is meticulously categorized; for instance, the classic GSM8K (Cobbe et al., 2021) dataset represents the math scenario of English, math word problems, and primary school tasks. To tackle the adaptation challenge, MathEval employs tailored prompts suitable for various models and datasets, ranging from zero-shot to few-shot settings. This ensures a thorough assessment of each model's capabilities across diverse problem sets, promoting a fair comparison of mathematical abilities across models. MathEval leverages GPT-4 for both answer extraction and comparison, thereby avoiding the complexities of regular expression rules and setting a consistent evaluation standard. We have validated GPT-4's effectiveness by comparing its outputs against human-annotated result, with only minimal discrepancies noted. To our knowledge, this is the first comprehensive benchmark specifically designed to evaluate the mathematical capabilities of LLMs holistically. We have evaluated 52 models across 22 datasets under varied adaptation conditions, making the results publicly accessible .

Contributions of MathEval are outlined as follows:

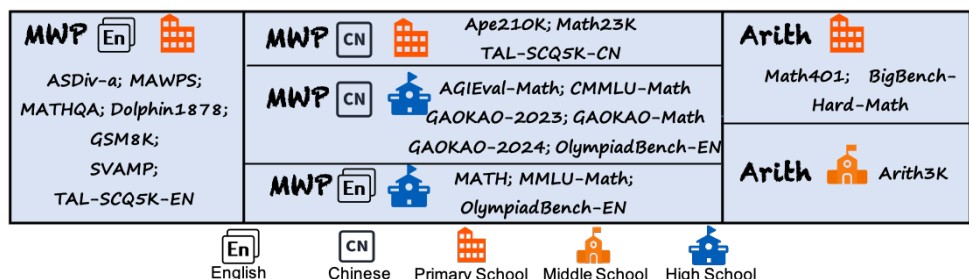

Figure 2: Overview of 22 datasets used in MathEval Framework. The datasets are categorized across three dimensions: problem type (Arithmetic - Arith, Math Word Problems - MWP), language (Chinese - CN, English - EN), and educational level (Primary - P, Middle - M, High - H). This organization ensures comprehensive coverage of various math scenarios for robust evaluation.

- MathEval provides an extensive benchmark that includes a diverse array of mathematical problems across different types and difficulty levels. This thorough categorization facilitates detailed analyses that can unveil new insights and directions for future research in the field of LLMs and mathematical reasoning.

- We have developed a standardized method for comparing answers that effectively addresses the complexities associated with outputs from mathematical word problems (MWPs). For broader accessibility, we also offer a self-developed compare-answer model for researchers and developers who do not have access to GPT-4.

- Recognizing the potential for data contamination in LLM evaluations, MathEval implements a strategy of using a dynamically updated dataset. This approach ensures that the evaluation reflects the true, unlearned capabilities of LLMs in solving mathematical problems, providing a more accurate measure of their realistic mathematical reasoning ability.

## 2 MATHEVAL

In this section, we will delve into the essential aspects of MathEval's implementation by elaborating on its three main components: math scenarios, prompt adaptations, and evaluation methods. Finally, we will introduce the entire pipeline to provide a comprehensive understanding of how these components integrate to form MathEval.

### 2.1 MATH SCENARIOS

Figure 2 presents MathEval's compilation of 22 math datasets utilized in leading conference papers since 2010, spanning six scenarios across problem types (arithmetic, math word problems), languages (Chinese, English), and educational levels (primary to high school). Notably, MathEval uniquely features the Arith3K, GAOKAO-2023, GAOKAO-2024, TAL-SCQ5K-EN, and TAL-SCQ5K-CN datasets, which are new additions not previously included in other benchmarks. Specifically, within the problem type dimension of our MathEval benchmark, three datasets—Arith3K, Big-Bench-Hard-Math, and Math401—focus solely on arithmetic problems, while the remaining 19 datasets are dedicated to MWPs. For the language dimension, it is important to note that only MWPs' datasets require language categorization, and these are nearly evenly split with 10 in English and 9 in Chinese. Regarding the educational level, 12 datasets target primary school, 9 cater to high school, and only Arith3K is designed for middle school students. Predominantly, the datasets focus on primary school level English MWP, followed by primary and high school level Chinese MWP scenarios. Detailed information about each dataset is available in the appendix B.2.

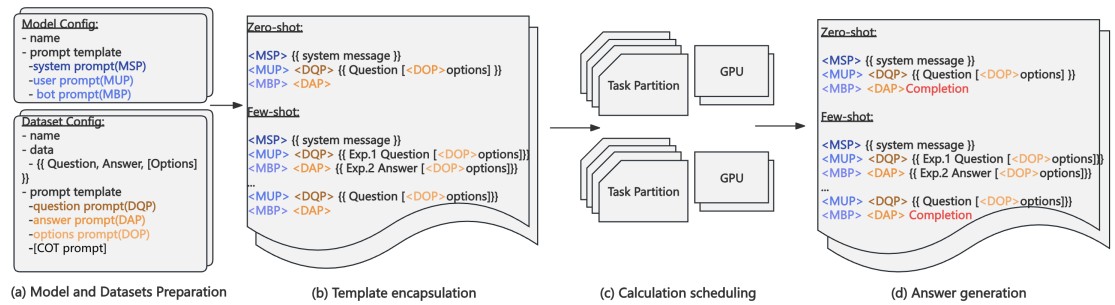

Figure 3: Four stages of the answer generation process: (a) model and datasets preparation, (b) template encapsulation, (c) computing scheduling and (d) Answer generation.

Building on this overview, we introduce some new datasets that have not been used by other benchmarks in MathEval: Arith3K, TAL-SCQ5K-EN, TAL-SCQ5K-CN, GAOKAO-2023 and GAOKAO-2024, each offering unique characteristics and challenges to the benchmark. For detailed descriptions of these datasets, please refer to the Appendix B.1.

## 2.2 PROMPTS ADAPTATION

As shown in Figure 3, the process involves four stages: model and datasets preparation, template encapsulation, computing scheduling, and answer generation.

**Model and Dataset Preparation:** This phase encompasses the establishment of model and dataset configurations for the ensuing stages. Users have the option to employ their own dataset and model configurations to expand the current benchmark or to evaluate their own models using MathEval. For the model configuration, the elements include: (1) Model name: The identifier for the model being used; (2) Prompt template: The general structure of prompts used by the model; (3) System prompt (MSP): The official system prompt from the model description or technical report; (4) User prompt (MUP): A token or phrase indicating the start or end of a user message; (5) Bot prompt (MBP): Similar to the user prompt, it indicates the start or end of a bot response; For the dataset configuration, the components consist of: (1) Data metadata: Information used to populate different parts of the template; (2) Question prompt (DQP): Indicates where the question is located and specifies the different types of questions; (3) Answer prompt (DAP): Specifies the kind of answer that needs to be generated, option from A to D or a specific answer; (4) Options prompt (DOP): Indicates where the options are located within the template; (5) Chain of Thought (CoT) prompt: Guides the model to output different types of CoT reasoning for each dataset. For example, in multiple-choice questions, the CoT should reason through each choice before providing the final answer; This detailed preparation ensures that both the model and datasets are configured correctly to facilitate accurate and contextually appropriate responses in the following stages. More details are discussed in Appendix H.1

**Template Encapsulation:** We encapsulate our final input prompt based on both model and dataset configurations. There are two scenarios: zero-shot prompt and few-shot prompt. Both settings use a combination of the previously discussed configuration elements. We include these two scenarios because base LLM models are generally not proficient in zero-shot scenarios, as they tend to continue generating content beyond the desired response. Introducing few-shot examples allows for a fair comparison by providing context and examples, thereby guiding the model to generate more accurate and contextually appropriate answers.

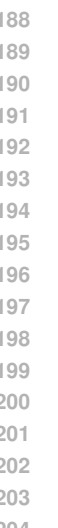
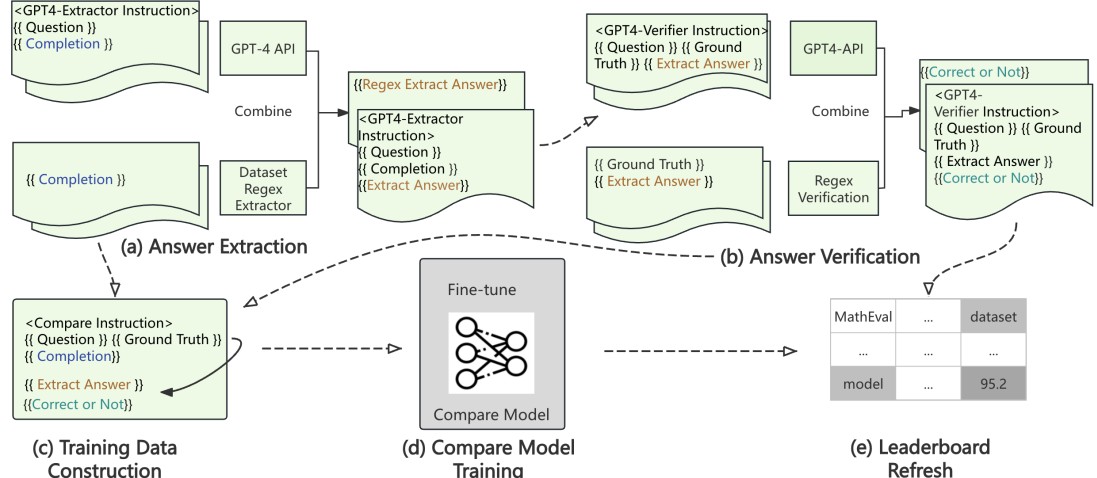

Figure 4: Two evaluation methods. (a)(b)(e) depict a two-stage method involving answer extraction and verification using GPT-4. (c)(d)(e) illustrate the training data construction of the comparison model and its training process.

**Calculation Scheduling and Answer Generation:** The final two stages of our methodology are Calculation Scheduling and Answer Generation[1]. At Calculation Scheduling stage, the task is automatically partitioned based on the available computational resources. This partitioning also takes into account the model size and dataset size, ensuring an efficient parallel processing to expedite the inference stage. Ultimately, this results in the generation of completion answers for both zero-shot and few-shot scenarios.

## 2.3 EVALUATION METHODS

The conventional metric for evaluation entails designing specific answer extraction rules tailored to the models and datasets, followed by a matching process. While this traditional approach can yield stable results, it suffers from a lack of robustness. Minor variations in the output can lead to significantly different outcomes. Moreover, crafting answer extraction rules for each model and dataset based on their output formats introduces quadratic complexity, making rule-based evaluation criteria inefficient. Consequently, we adopt a general evaluation method that can be easily and cost-effectively extended to new datasets and models, thereby enhancing the fairness of the evaluation process through a unified standard.

We initially employed a two-stage evaluation method, as depicted in Figure 4a and 4b. In the first stage, answer extraction, the generated response is processed to isolate the specific answer. In the subsequent stage, answer verification, the extracted answer is compared against the ground truth to produce a comparison result. Given the robustness of methods based on LLMs, such as GPT-4 (OpenAI, 2023), these models exhibit strong comprehension capabilities and can handle diverse output formats. In contrast, rule-based methods offer greater stability in obtaining results. Consequently, in both evaluation pipelines, we primarily utilize outputs from GPT-4, supplementing them with regex-based results when GPT-4 fails. Detailed instructions for answer extraction and verification using GPT-4 are provided in Appendix F. Additionally, a comprehensive

---

[1]Please refer code in `https://anonymous.4open.science/r/MathEval-505B/README.md`

comparison between the regex-rule-based method and our GPT-4-as-judgement method is presented in Appendix F.4.

Subsequently, as illustrated in Figure 4c and 4d, we developed an answer comparison model that takes as input the question, the model-generated answer, and the reference (golden) answer, and outputs a detailed, step-by-step analysis to extract and assess the correctness of the generated answers. This process represents a holistic, end-to-end evaluation activity. By consolidating what was previously a bifurcated process, this method enhances both the stability and cost-efficiency of the evaluation approach. An exemplar of the training data utilized is provided in Appendix G.1. We collected a total of 2,217,328 evaluation results derived from GPT-4 under the former two-stage paradigm, which served as the training dataset for our DeepSeek-7b (Shao et al., 2024) based answer comparison model. While both methodologies offer distinct advantages, we predominantly employ the initial method—the two-stage process leveraging GPT-4—as our principal source of results.

## 3 EXPERIMENT

### 3.1 EVALUATED MODELS

52 different models have been evaluated, we have categorized these models into three distinct groups. The first group consists of open-source models, characterized by their accessibility in terms of model weights and architecture. The second group comprises closed-source models, which are accessible only through APIs without disclosure of their underlying architecture or weights. The third category specifically includes open-source models that have been fine-tuned on math domain data, allowing for a more tailored analysis in this specific area. Models within each categorized group listed in Appendix C.

### 3.2 COMPARE ANSWER MODELS

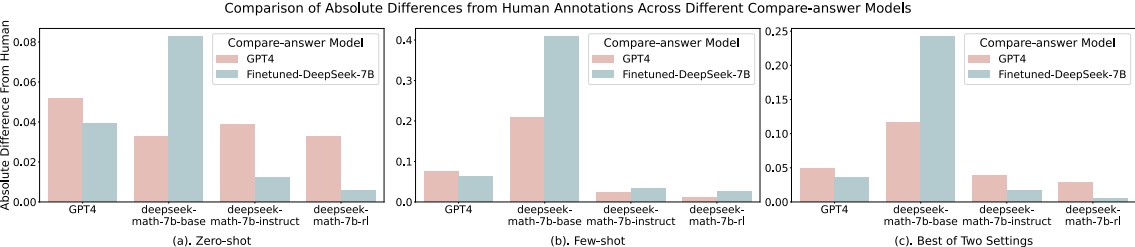

Figure 5: Comparison of Absolute Differences from Human Answers Across Different Compare-answer Models.

In this section, we explore the methodologies selected for the compare-answer task, a notably challenging aspect of mathematical benchmark. This complexity is well-acknowledged within the academic community, as highlighted in several case studies included in the Appendix G.2. Many benchmarks in this field also encounter difficulties due to their reliance on rule-based extraction and matching approaches. These methods typically struggle to accommodate the diverse output behaviors exhibited by different models. Furthermore, there has been a lack of focused research on this domain, particularly in terms of a systematic analysis of the accuracy of answer matching across various models. This section aims to address these gaps by detailing our approach and the rationale behind our methodological choices.

To authenticate the precision of our methodology, we have organized a large-scale human annotation process of the model output results, which was carried out over the course of approximately one month. We have

selected GPT4, DeepSeek-Math-7B-Base, DeepSeek-Math-7B-Instruct, and DeepSeek-Math-7B-RL as the basis for validation. Five annotators were assigned to annotate each line of outputs, and the majority vote result was considered the final decision on the correctness of the model output. The details of the data can be found in Appendix G.3. The overall Fleiss' Kappa (Fleiss, 1971) achieved a score of 0.8871, indicating significant inter-annotator agreement. We believe the human annotation result is reliable and treat the overall average accuracy as the golden standard.

We evaluated two methods for answer comparison. The first is our two-stage GPT4-based judgment depicted in Section 2.3. The second is a fine-tuned DeepSeek-7B-Base model (Finetuned-DeepSeek) trained on our private answer-comparison data and GPT4 output comparison data (partially verified by human annotators to fix potential errors). We computed the overall average accuracy for each answer-comparison method using 19 selected datasets out of 22 from the four chosen models. The results are shown in Figure 5, where the y-axis represents the absolute difference between the proposed answer-comparison method and the human evaluation result, with larger values indicating worse performance. We present the zero-shot, few-shot, and best-of-two previous settings in this figure, focusing primarily on the best-of-two setting. Initially, we observed that GPT4 performed consistently well across all models, with an absolute difference ranging from 0 to 0.1. Both methods performed poorly on the output of DeepSeek-Math-7B-Base, likely due to the base model's tendency to output useless tokens and inability to stop at the correct position, which poses challenges for the answer-comparison model. Notably, Finetuned-DeepSeek achieved the same performance as human annotators on the output of DeepSeek-Math-7B-RL, demonstrating the effectiveness of our method. Given GPT4's consistent performance, it will be our primary model for further analysis. We have open-sourced our custom Finetuned-DeepSeek model to provide a viable alternative for those without access to GPT-4.

## 3.3 EVALUATION RESULTS

The main results of MathEval are detailed in Table 1, comprehensive results for additional models can be found in Appendix D. We calculated the arithmetic mean accuracy for each model across 22 datasets and ranked the models within each group based on this metric, which we refer to as the overall average. In the subsequent analysis, the overall average is predominantly used as the primary metric for evaluation. To ensure the credibility of our evaluation results, as detailed in Appendix F.3, we compared our evaluated results from GPT-4 with the reported metrics of each published model on the GSM8K and MATH datasets, which are commonly used for assessing math-solving abilities. The minor discrepancies observed demonstrate the reliability of our evaluation pipeline.

**Top1 in 3 categories models.** Claude-3.5-Sonnet, a closed-source model, has demonstrated exceptional performance, surpassing GPT-4 by a significant margin with an average accuracy of 77.0%. This superiority is evident across various dimensions, particularly in its advanced understanding of both English and Chinese languages. Claude-3.5-Sonnet's proficiency in handling high school level problems further highlights its reasoning capabilities. For open-source models, Qwen2-72B-Instruct leads the pack with an impressive average accuracy of 74.4%. This model's performance is followed closely by Qwen15-110B-Chat, which also shows strong results, indicating that the newer large-parameter chat models possess superior mathematical abilities. In the math domain, despite having only 7 billion parameters, deepseek-math-7b-rl stands out with an average accuracy of 63.0%, showcasing the effectiveness of its post-training. However, it still trails behind the top-5 open-source models, which are all outperformed by the top-3 closed-source models. This underscores the importance of model parameter size in achieving leading mathematical capabilities and highlights the current gap between open-source and closed-source models.

## 3.4 DISCUSSION

With MathEval, we have uncovered several intriguing insights. We will delve into these findings in detail within this section.

Table 1: Summary of principal outcomes from MathEval. Abbreviations used: 'En.' for English, 'Cn.' for Chinese, 'Arith.' for Arithmetic, 'Prim' for Primary, 'Mid.' for Middle, and 'Avg' for average score. The table displays the top-six performing models in each category.

| Models | Language | | | Type | | Grade | | Avg. |
|---|---|---|---|---|---|---|---|---|
| | En. | Cn. | MWP | Arith. | Prim. | Mid. | High | |
| *Closed-source Models* | | | | | | | | |
| Claude-3.5-Sonnet | 84.7 | 67.2 | 76.4 | 80.8 | 90.0 | 57.3 | 61.8 | 77.0 |
| WenXin 4.0 | 78.3 | 65.7 | 72.4 | 93.1 | 88.2 | 89.6 | 56.3 | 75.2 |
| Gemini1.5Pro | 81.9 | 63.8 | 73.3 | 81.9 | 88.8 | 46.9 | 58.5 | 74.5 |
| GLM4 | 76.5 | 61.3 | 69.3 | 60.9 | 83.1 | 32.4 | 52.2 | 68.1 |
| Spark-3.5 | 72.8 | 60.6 | 67.0 | 68.4 | 81.5 | 41.2 | 51.1 | 67.2 |
| GPT4 | 72.4 | 45.9 | 59.8 | 67.1 | 79.6 | 38.3 | 38.3 | 60.8 |
| *Open-source Models* | | | | | | | | |
| Qwen2-72B-Instruct | 81.8 | 64.7 | 73.7 | 78.7 | 88.7 | 57.3 | 57.2 | 74.4 |
| Qwen15-110B-Chat | 76.3 | 57.3 | 67.3 | 68.6 | 84.0 | 40.8 | 48.4 | 67.5 |
| Qwen2-72B | 73.0 | 57.0 | 65.4 | 65.4 | 79.7 | 35.3 | 49.7 | 65.4 |
| LLama-3-70B-Instruct | 76.6 | 51.7 | 64.8 | 68.8 | 82.3 | 42.4 | 45.3 | 65.4 |
| Qwen2-7B-Instruct | 75.8 | 52.7 | 64.8 | 67.4 | 81.3 | 46.3 | 45.8 | 65.2 |
| Qwen15-72B-Chat | 71.7 | 55.1 | 63.8 | 62.8 | 79.6 | 33.4 | 45.7 | 63.7 |
| *Math Domain Models* | | | | | | | | |
| deepseek-math-7b-rl | 74.0 | 50.3 | 62.8 | 64.4 | 79.5 | 44.0 | 43.0 | 63.0 |
| deepseek-math-7b-instruct | 69.7 | 46.7 | 58.8 | 57.7 | 75.7 | 36.6 | 38.3 | 58.7 |
| internlm2-math-20b | 66.0 | 44.7 | 55.9 | 41.3 | 68.4 | 28.8 | 37.4 | 53.9 |
| MetaMath-70B | 57.6 | 27.7 | 43.4 | 32.1 | 58.3 | 12.5 | 23.3 | 41.9 |
| MAmmoTH-70B | 56.5 | 27.6 | 42.8 | 30.9 | 56.6 | 11.4 | 23.9 | 41.2 |
| GAIRMath-Abel-70B | 53.5 | 30.8 | 42.7 | 25.5 | 53.3 | 11.5 | 26.3 | 40.4 |

**Closed-source models** exhibit a higher capability range than open-source models and math domain models. As shown in the Figure 6a, not only does it exhibit the highest capability ceiling, but it also maintains a high capability floor, with only GPT-3.5 lagging slightly. This indicates that closed-source models typically exhibit consistently superior performance in mathematical tasks. Nevertheless, we also observed that the 25th percentile range of closed-source models is encompassed by the capability range of open-source models. This suggests that excellent open-source models can achieve performance comparable to closed-source models.

**Open-source models** exhibit a wide range of capabilities influenced by both the type of base model and the size of the model parameters, as shown in Figure 6b. While the size of the model parameters does not directly determine the model's mathematical abilities, it can increase the potential upper limit of these abilities. Consistent with general conclusions, we observe that the mathematical ability of models with the same base architecture has a linear relationship with the logarithm of their parameter sizes. Additionally, chat models consistently outperform base models, reflecting the stabilizing effect of post-training. Furthermore, analyzing the lines of similar color in the figure reveals that the slopes of models with the same base architecture are remarkably uniform. Interestingly, newer series exhibit steeper slopes, indicating that their mathematical abilities improve more effectively with an increase in parameter size.

**For problem type dimension**, as shown in Figure 6d, the scarcity of arithmetic-related datasets leads to significant fluctuations in arithmetic capabilities across models, represented by the blue line. Models highlighted in blue, positioned below the average difference line, exhibit stronger arithmetic abilities compared to their capabilities in solving MWPs. For closed-source models, the notable deviations of WenXin 4.0 may be due to their arithmetic plugins. We did not use API versions with plugins for GPT-4, which could affect their performance in arithmetic tasks. Other open-source models like LLama-3-70B and internlm2-base-20b also show strong arithmetic capabilities. Conversely, models above the average difference line, highlighted in red, are predominantly fine-tuned on MWPs data. This includes specialized models such as MAmmoTH-70B,

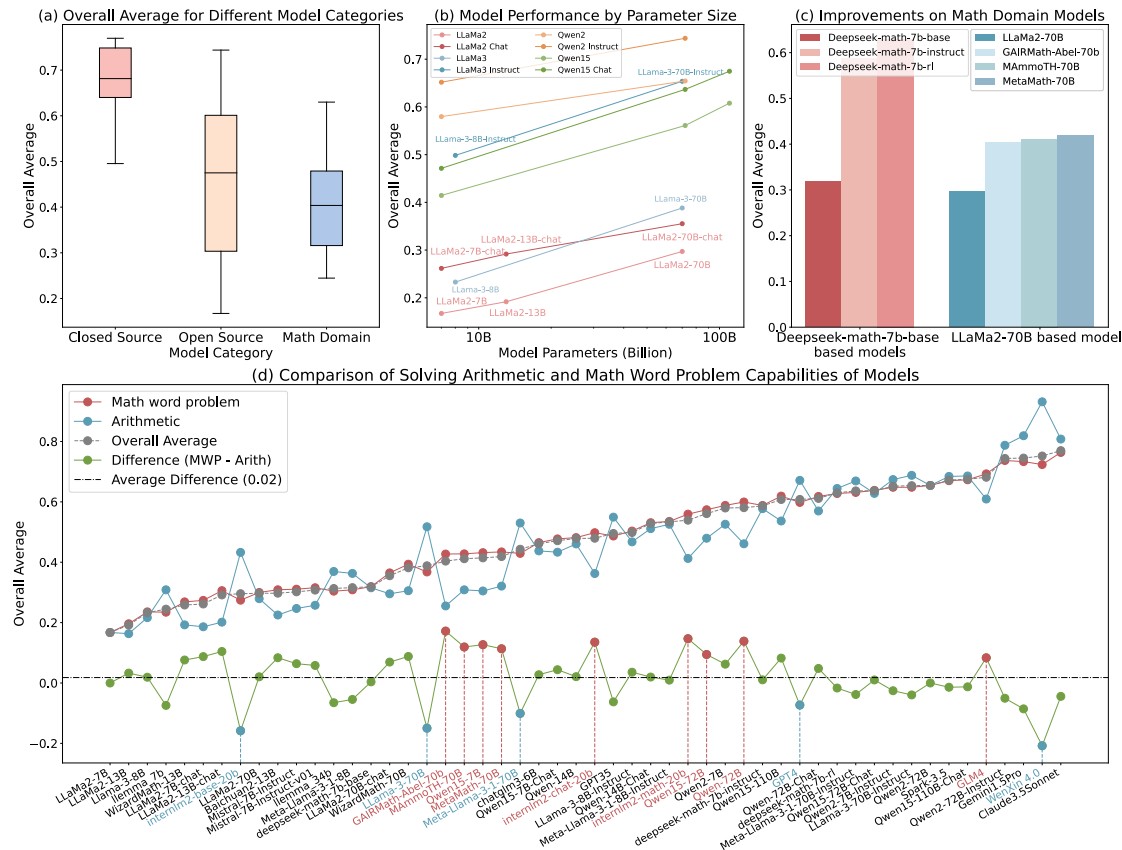

Figure 6: MathEval evaluation results. (a)-(c) show the discovery of closed-source models, open-source models, and math domain models. (d) compares the model-level capabilities across problem type dimensions.

MetaMath-70B, and GAIRMath-Abel-70b. A similar trend is evident in internlm2-math-20b and internlm2-chat-20b, which, unlike the more arithmetic-proficient internlm2-base-20b, likely benefitted from targeted fine-tuning on MWPs datasets during the post-training phase. Additionally, models like Qwen1.5-72B, Qwen-72B, and GLM4 also demonstrate enhanced capabilities in handling MWPs.

**Few-shot/zero-shot setting** is a relatively consistent part of prompts adaptation. We aim to maintain consistency while ensuring fairness, making it important to understand how different few-shot/zero-shot settings affect model capabilities. Specifically, our evaluation includes three settings: few-shot, zero-shot, and the higher accuracy between few-shot and zero-shot at the dataset level. As shown in the Figure 7, the "dataset-level higher" setting consistently outperforms using either few-shot or zero-shot alone across all models. It also produces smoother curves with fewer outliers, indicating that this setting contributes to the robustness and fairness of the evaluation. When comparing zero-shot and few-shot, zero-shot generally performs better on most models, with only some base models showing significantly lower performance (represented by red dashed lines). Notably, the base models in the Qwen series do not exhibit this phenomenon.

We conduct more discussions in Apendix E.

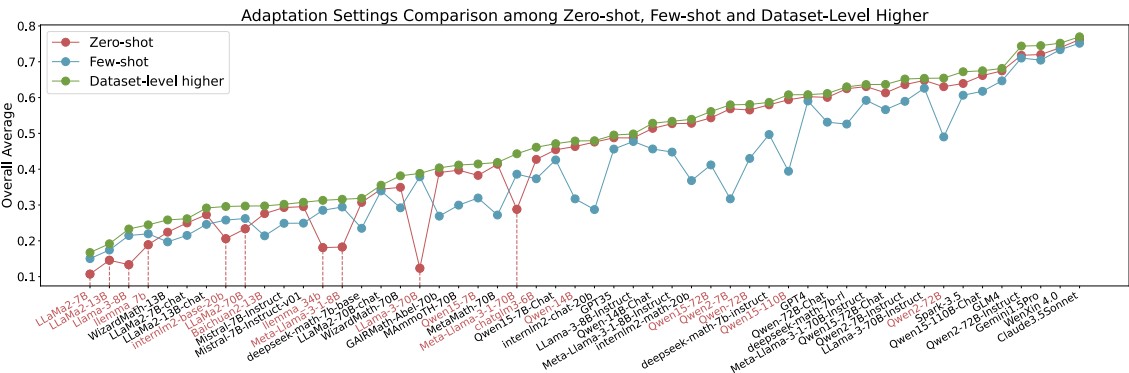

Figure 7: Comparison of prompts adaptation settings: few-shot, zero-shot and datset-level higher. Base models are highlighted in red on the x-axis, while post-training models are shown in black.

## 4 RELATED WORK

Benchmarks orient LLMs. **General benchmarks** provide a comprehensive evaluation of LLMs and are widely used across various natural language understanding tasks to assess their performance. The **MMLU** (Hendrycks et al., 2021a) benchmark is notable for its extensive collection of 57 tasks covering diverse domains, offering comprehensive challenges across varying subjects and levels of complexity. **AGIEval** (Zhong et al., 2023) is centered on standardized exams like the SAT, LSAT, and GRE, testing models' reasoning, problem-solving, and language comprehension skills. The broader **BIG-Bench** (Srivastava et al., 2022) initiative includes a diverse set of tasks designed to probe models on novel and complex linguistic capabilities, challenging them to demonstrate their robustness and versatility in a wide array of cognitive tasks beyond traditional benchmarks.

Domain-specific benchmarks are crucial for evaluating how well LLMs handle specialized tasks requiring deep field knowledge. **HaluEval** (Li et al., 2023) assesses hallucination detection in LLMs using annotated samples, revealing that models frequently generate unverifiable information. **LongBench** (Bai et al., 2023b) tests long-context comprehension in English and Chinese across 21 datasets, showing that expanding context windows and enhancing memory mechanisms improve long-sequence understanding.

To the best of our knowledge, there is currently no comprehensive mathematical evaluation benchmark. A similar mathematical benchmark, **Lila** (Mishra et al., 2023), focuses on extending datasets by collecting task instructions and solutions as Python programs and then exploring some models' out-of-domain capabilities. A comprehensive benchmark for assessing the mathematical abilities of various models remains absent.

## 5 CONCLUSION

In this paper, we proposed **MathEval**, the first comprehensive evaluation benchmark for the mathematical capabilities of large language models (LLMs). Our evaluation encompassed 52 models across 22 datasets, organized into distinct scenarios along three dimensions. Our pipeline facilitates flexible adaptation to various datasets and models. Moreover, we propose an LLM-based approach for the automatic extraction and verification of mathematical answers, serving as a general and precise metric. We hope that MathEval will help provide an impartial evaluation of the mathematical abilities of LLMs, advancing the continuous improvement of LLM mathematical capabilities and expanding practical applications.

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

# A APPENDIX

# B DATASETS DETAILS

## B.1 UNIQUE DATASET DESCRIPTION

Arith3K is a high quality arithmetic evaluation dataset we constructed, consists of 3 main categories and 15 sub-categories, totaling 3,000 problems. It includes 12 types of mathematical operations, ranging from simple arithmetic and logarithmic operations to factorials, trigonometric functions, and complex compound operations. We systematically combined numbers and operators, and used Python code along with SymPy to verify the correctness of each expression. This makes Arith3K the most challenging dataset among arithmetic collections in our benchmark, designed to comprehensively assess the computational abilities of LLMs across varied difficulty levels.

TAL-SCQ5K-EN and TAL-SCQ5K-CN are comprehensive mathematical competition datasets available in English and Chinese, respectively. Each dataset comprises 5,000 multiple-choice questions, divided into 3,000 for training and 2,000 for testing, covering primary, junior high, and high school level mathematics. These datasets are particularly valuable for Chain of Thought (CoT) training as they include detailed solution steps. Furthermore, all mathematical expressions within the questions are formatted in standard text-mode LaTeX, ensuring clarity and consistency in presentation. To maintain the high quality of the TAL-SCQ datasets, each question undergoes a rigorous review process by two qualified teachers before being included. Moreover, an independent quality validation was conducted on a randomly selected sample of 200 problems, all of which were approved by independent teachers, with no identified issues.

The GAOKAO-2023 and GAOKAO-2024 datasets are derived from the most recent Chinese National College Entrance Examination and consist of both multiple-choice and mathematical word problems. These datasets, which reflect actual exam content, will be updated on an annual basis with forthcoming versions such as GAOKAO-2025[2]. These consistent updates are designed to help alleviate potential contamination of test data.

We focus on K-12 education levels due to their broad applicability and the availability of extensive datasets. However, we recognize that incorporating higher-level mathematics—such as undergraduate topics and competition math problems like PutnamBench—would provide deeper insights into the models' capabilities across varying difficulty levels. We are actively working to include these more challenging problems in future iterations of MathEval.

---

[2]Dataset will be updated annualy in our Github

## B.2 DATASET CATEGORIZATION AND DETAILED PROBLEM ANALYSIS

In this section, we conduct a detailed analysis of the differences among the datasets and our categorization of them to avoid the issue of measuring the same ability dimensions. First, in Table 2, we present each of our datasets and their corresponding classifications, including language, problem type, and corresponding grade level (which can partially reflect the difficulty level).

To further analyze the distinctions between the datasets, we examined the data distribution of problems in the different datasets to ensure that they are dissimilar. For each dataset, we first randomly sampled 200 query problems and obtained their representations using the `LLaMa-3-8B` Dubey et al. (2024) model. We performed t-SNE dimensionality reduction on these representations, with the visualization shown in Figure 9 and the cosine similarity situation in Figure 8.

For the calculation of cosine similarity, we computed the cosine similarity between each query across every pair of datasets and finally took the absolute value of the average as their correlation. We found that the correlation scale between dataset queries ranges from 0 to 0.8, and according to statistics, 75.32% are less than 0.6, and 60.17% are less than 0.5. This demonstrates the dissimilarity between dataset queries, reflecting that to some extent they measure different abilities.

Moreover, we made a surprising discovery in the t-SNE results: the t-SNE naturally formed three clusters. One cluster consists of English datasets (SVAMP, GSM8K, MathQA, MATH, Dolphin-1878, OlympiadBench-EN, etc.), another cluster consists of Chinese datasets (CMMLU-Math, Math23K, Ape210K, TAL-SCQ5K-CN, etc.), and a third cluster consists of arithmetic problems (Big-Bench-Hard-Math, Math401, Arith3K). From component 1, we can observe that on the left are the English datasets, in the middle are the arithmetic problems, and on the right are the Chinese datasets.

Further observing the Chinese and English clusters, we find that as the t-SNE component 2 value decreases (from top to bottom in the figure), the problems become increasingly difficult, and the corresponding grade levels also rise. This perfectly reflects that our difficulty levels are distributed across various grades, with corresponding datasets at each level. This shows that our 22 datasets measure different aspects, whether in terms of difficulty or language.

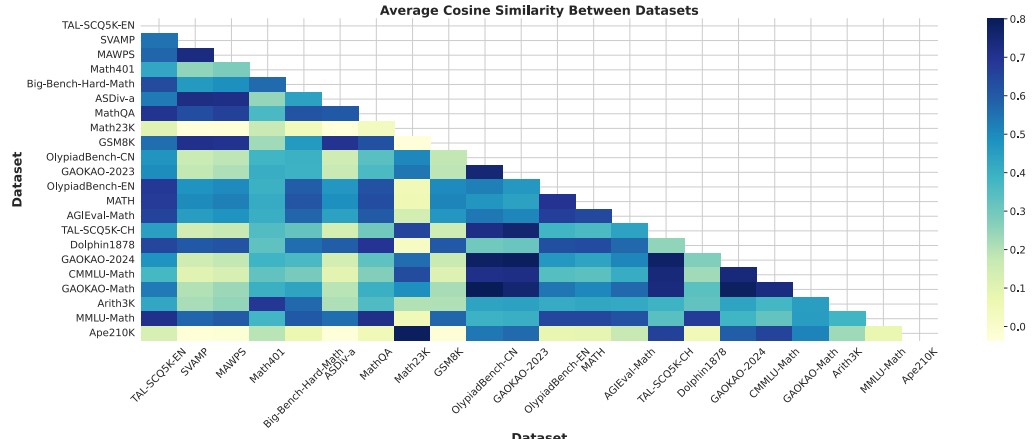

Figure 8: Average cosine similarity of query embeddings between each pair of the 22 datasets.

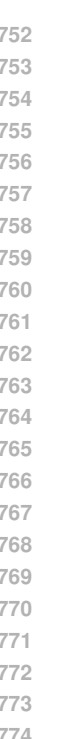

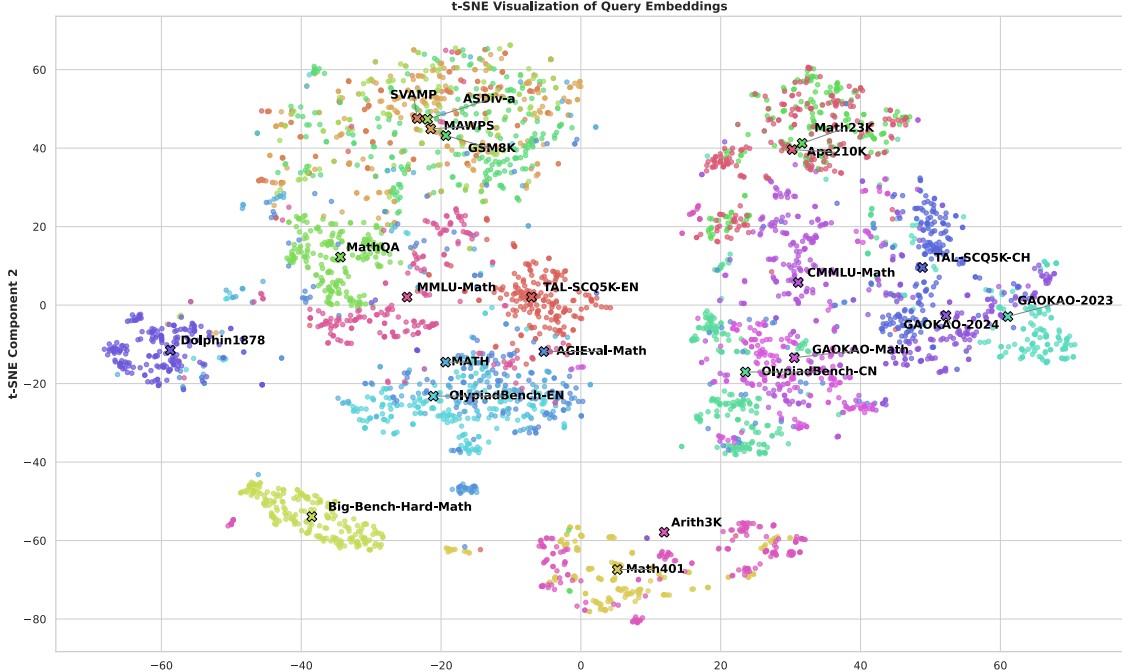

Figure 9: t-SNE visualization of query embeddings from the 22 datasets.

## C  MODEL CATEGORIES AND SOURCES

Closed-source models: GPT-4, GPT-3.5[3], GLM4[4], WenXin 4.0[5], Spark-3.5[6], Gemini-1.5-Pro[7], Claude-3.5-Sonnet[8]

Open-source models: Qwen, Qwen1.5 and Qwen2 (Bai et al., 2023a);(Yang et al., 2024), LLaMa2, LLaMa3 and LLaMa3.1 (Touvron et al., 2023);(Dubey et al., 2024), Mistral (Jiang et al., 2023), InternLM2 (Cai et al., 2024), Moss-Moon (Sun et al., 2023), Baichuan2 (Yang et al., 2023), ChatGLM3 Du et al. (2022)

Math domain models: WizardMath (Luo et al., 2023), MAmmoTH (Yue et al., 2023), MetaMath (Yu et al., 2023), Llemma (Azerbayev et al., 2023), GAIRMath-Abel (Chern et al., 2023), Deepseek-Math (Shao et al., 2024)

---

[3]GPT-4 & GPT-3.5 version 2024-02-01
[4]https://open.bigmodel.cn
[5]https://cloud.baidu.com/qianfandev
[6]https://xinghuo.xfyun.cn/spark
[7]https://deepmind.google/technologies/gemini/pro/
[8]https://www.anthropic.com/news/claude-3-5-sonnet

Table 2: 22 datasets used in MathEval, along with their references and three-dimensional categories.

| Model | Language | Problem Type | Grade |
|---|---|---|---|
| **AGIEval-Math** (Zhong et al., 2023) | Chinese | Math word problem | High School |
| **Ape210K** (Zhao et al., 2020) | Chinese | Math word problem | Primary School |
| **Arith3K** (Ours) | - | Arithmetic | Middle School |
| **ASDiv-a** (Miao et al., 2020) | English | Math word problem | Primary School |
| **Big-Bench-Hard-Math** (Suzgun et al., 2022) | - | Arithmetic | Primary School |
| **CMMLU-Math** (Li et al., 2024) | Chinese | Math word problem | High School |
| **Dolphin1878** (Shi et al., 2015) | English | Math word problem | Primary School |
| **GAOKAO-2023** (Ours) | Chinese | Math word problem | High School |
| **GAOKAO-2024** (Ours) | Chinese | Math word problem | High School |
| **GAOKAO-Math** (Zhang et al., 2024) | Chinese | Math word problem | High School |
| **GSM8K** (Cobbe et al., 2021) | English | Math word problem | Primary School |
| **MAWPS** (Koncel-Kedziorski et al., 2016) | English | Math word problem | Primary School |
| **Math23K** (Wang et al., 2017) | Chinese | Math word problem | Primary School |
| **Math401** (Yuan et al., 2023) | - | Arithmetic | Primary School |
| **MATH** (Hendrycks et al., 2021b) | English | Math word problem | High School |
| **MATHQA** (Amini et al., 2019) | English | Math word problem | Primary School |
| **MMLU-Math** (Hendrycks et al., 2021a) | English | Math word problem | High School |
| **TAL-SCQ5K-CN** (Ours) | Chinese | Math word problem | Primary School |
| **TAL-SCQ5K-EN** (Ours) | English | Math word problem | Primary School |
| **SVAMP** (Patel et al., 2021) | English | Math word problem | Primary School |
| **OlympiadBench-CN** (He et al., 2024) | Chinese | Math word problem | High School |
| **OlympiadBench-EN** (He et al., 2024) | English | Math word problem | High School |

## D   MATHEVAL MAIN RESULT

Due to space constraints, the detailed performance of each dataset for every model are not included in this paper. However, these details will be made available upon the paper's acceptance. We have summarized the

average results across each domain and the total arithmetic average scores in Table 3 and Table 4. All models are ranked by their arithmetic average within each source, and a comprehensive final ranking across all categories is provided in Table 5. Abbreviations used: 'En.' for English, 'Cn.' for Chinese, 'MWP' for Math word problem, 'Arith.' for Arithmetic, 'Prim' for Grade School, 'Mid.' for Middle School, 'High' for High School, and 'Avg' for overall average score. We also calculated the Pearson correlation coefficients between datasets based on the evaluation results of the models, reflecting the linear relationships among the datasets, as shown in Figure 10. The Pearson correlation coefficients between datasets are relatively high because we are assessing mathematical abilities, which are inherently strongly correlated. If a model's computational ability improves, its problem-solving ability will naturally improve. Similarly, if a model's accuracy increases on high school-level questions, its success rates on easier middle school and elementary school problems will also be higher. The high Pearson correlations we observed demonstrate the robustness of our benchmark.

Nevertheless, we still selected 22 datasets for two main reasons. First, we want our test results to be more robust; only with as much data as possible can we truly reflect the actual performance of the models, and potentially discover any poor performance on certain datasets. Second, through the query analysis in the previous section, we found that our 22 datasets cover different languages and difficulty levels, ensuring a balanced and comprehensive representation across these dimensions.

Table 3: Final ranking for closed-source and math domain models from MathEval.

| Models | Language | | | Type | | Grade | | | Avg. |
|---|---|---|---|---|---|---|---|---|---|
| | En. | Ch. | MWP | Arith. | Prim. | Mid. | High | | |
| *Closed-source Models* | | | | | | | | | |
| Claude-3.5-Sonnet | 84.7 | 67.2 | 76.4 | 80.8 | 89.9 | 57.3 | 61.8 | | 77.0 |
| WenXin 4.0 | 78.3 | 65.7 | 72.4 | 93.1 | 88.2 | 89.6 | 56.3 | | 75.2 |
| Gemini-1.5-Pro | 81.9 | 63.8 | 73.3 | 81.9 | 88.8 | 46.9 | 58.5 | | 74.5 |
| GLM4 | 76.5 | 61.3 | 69.3 | 60.9 | 83.1 | 32.4 | 52.2 | | 68.1 |
| Spark-3.5 | 72.8 | 60.6 | 67.0 | 68.4 | 81.5 | 41.2 | 51.1 | | 67.2 |
| GPT-4 | 72.4 | 45.9 | 59.8 | 67.1 | 79.6 | 38.3 | 38.3 | | 60.8 |
| GPT-3.5 | 61.2 | 34.8 | 48.7 | 54.9 | 66.7 | 35.4 | 28.2 | | 49.5 |
| *Math Domain Models* | | | | | | | | | |
| Deepseek-Math-7B-RL | 74.0 | 50.3 | 62.8 | 64.4 | 79.5 | 44.0 | 43.0 | | 63.0 |
| Deepseek-Math-7B-Instruct | 69.7 | 46.7 | 58.8 | 57.7 | 75.7 | 36.6 | 38.3 | | 58.7 |
| InternLM2-Math-20B | 66.0 | 44.7 | 55.9 | 41.3 | 68.4 | 28.8 | 37.4 | | 53.9 |
| MetaMath-70B | 57.6 | 27.7 | 43.4 | 32.1 | 58.3 | 12.5 | 23.3 | | 41.9 |
| MAmmoTH-70B | 56.5 | 27.6 | 42.8 | 30.9 | 56.6 | 11.4 | 23.9 | | 41.2 |
| GAIRMath-Abel-70B | 53.5 | 30.8 | 42.7 | 25.5 | 53.3 | 11.5 | 26.3 | | 40.4 |
| WizardMath-70B | 50.3 | 27.2 | 39.4 | 30.6 | 51.3 | 12.6 | 23.5 | | 38.2 |
| Deepseek-Math-7B-base | 36.0 | 27.4 | 31.9 | 31.5 | 39.8 | 21.7 | 22.4 | | 31.9 |
| Llemma-34B | 37.6 | 22.4 | 30.4 | 36.9 | 38.5 | 13.7 | 23.7 | | 31.3 |
| WizardMath-13B | 35.3 | 17.6 | 26.9 | 19.3 | 34.3 | 7.1 | 16.6 | | 25.8 |
| Llemm-7B | 27.5 | 19.0 | 23.4 | 30.9 | 29.5 | 8.9 | 19.4 | | 24.5 |

Table 4: Final ranking for open-source models from MathEval.

| Models | Language | | Type | | | Grade | | Avg. |
|---|---|---|---|---|---|---|---|---|
| | En. | Ch. | MWP | Arith. | Prim. | Mid. | High | |
| *Open-source Models* | | | | | | | | |
| Qwen2-72B-Instruct | 77.6 | 63.0 | 70.7 | 73.1 | 85.1 | 40.4 | 55.6 | 71.0 |
| LlaMa-3-70B-Instruct | 73.4 | 49.0 | 61.8 | 67.5 | 79.3 | 40.3 | 42.8 | 62.6 |
| Qwen15-110B-Chat | 66.1 | 55.1 | 60.9 | 67.0 | 75.9 | 35.9 | 45.8 | 61.8 |
| LlaMa-3.1-70B-Instruct | 72.0 | 42.7 | 58.1 | 66.1 | 75.5 | 35.5 | 40.1 | 59.2 |
| Qwen2-7B-Instruct | 66.4 | 49.9 | 58.6 | 61.1 | 73.4 | 30.4 | 42.8 | 58.9 |
| Qwen15-72B-Chat | 60.1 | 51.7 | 56.1 | 60.0 | 68.7 | 24.9 | 44.0 | 56.6 |
| Qwen-72B-Chat | 56.3 | 50.8 | 53.7 | 49.6 | 63.9 | 20.3 | 42.4 | 53.1 |
| Qwen2-72B | 56.9 | 34.8 | 46.5 | 65.2 | 61.1 | 35.3 | 34.3 | 49.0 |
| LlaMa-3-8B-Instruct | 60.9 | 33.5 | 47.9 | 46.5 | 64.2 | 22.6 | 28.5 | 47.7 |
| Qwen-14B-Chat | 50.3 | 39.9 | 45.4 | 47.3 | 57.7 | 19.7 | 32.4 | 45.6 |
| LlaMa-3.1-8B-Instruct | 59.0 | 26.6 | 43.6 | 52.2 | 60.6 | 24.1 | 26.0 | 44.8 |
| Qwen-72B | 42.1 | 43.2 | 42.6 | 45.3 | 49.9 | 14.6 | 36.9 | 43.0 |
| Qwen15-7B-Chat | 48.7 | 35.8 | 42.6 | 42.7 | 55.9 | 17.8 | 27.6 | 42.6 |
| Qwen15-72B | 38.3 | 42.5 | 40.3 | 46.9 | 49.0 | 12.8 | 34.0 | 41.2 |
| Qwen15-110B | 35.0 | 41.2 | 37.9 | 48.9 | 40.6 | 18.8 | 40.1 | 39.4 |
| LlaMa-3.1-70B | 47.4 | 24.0 | 36.3 | 53.0 | 52.4 | 22.2 | 22.0 | 38.6 |
| LlaMa-3-70B | 42.1 | 29.2 | 36.0 | 49.9 | 49.6 | 0.8 | 26.4 | 37.9 |
| ChatGLM3-6B | 42.8 | 29.4 | 36.5 | 42.9 | 47.4 | 13.0 | 26.6 | 37.3 |
| LlaMa2-70B-chat | 46.5 | 21.8 | 34.8 | 28.7 | 46.8 | 14.5 | 19.0 | 33.9 |
| Qwen15-7B | 35.1 | 29.9 | 32.6 | 27.6 | 42.0 | 17.9 | 20.1 | 31.9 |
| Qwen2-7B | 32.9 | 29.2 | 31.2 | 35.3 | 37.0 | 20.7 | 25.9 | 31.7 |
| Qwen-14B | 32.7 | 29.7 | 31.3 | 34.6 | 36.2 | 14.6 | 27.7 | 31.7 |
| LlaMa-3.1-8B | 37.5 | 18.2 | 28.4 | 36.3 | 40.9 | 12.5 | 16.0 | 29.4 |
| InternLM2-chat-20b | 34.0 | 25.4 | 29.9 | 21.4 | 34.7 | 11.1 | 22.7 | 28.7 |
| LlaMa2-70B | 33.3 | 18.1 | 26.1 | 26.9 | 33.5 | 11.0 | 18.3 | 26.2 |
| InternLM2-base-20b | 28.0 | 18.8 | 23.7 | 39.4 | 31.8 | 14.2 | 19.1 | 25.8 |
| Mistral-7B-Instruct-v01 | 34.2 | 14.4 | 24.8 | 25.7 | 33.2 | 11.9 | 15.3 | 24.9 |
| Mistral-7B-Instruct | 34.7 | 14.1 | 24.9 | 24.7 | 33.5 | 11.8 | 14.9 | 24.9 |
| LlaMa2-13B-chat | 32.4 | 17.5 | 25.3 | 20.0 | 31.8 | 10.0 | 16.6 | 24.6 |
| LlaMa2-7B-chat | 28.4 | 14.9 | 22.0 | 18.6 | 28.0 | 8.2 | 14.4 | 21.5 |
| LlaMa-3-8B | 26.4 | 15.9 | 21.5 | 21.6 | 28.7 | 9.4 | 13.1 | 21.5 |
| Baichuan2-13B | 25.6 | 16.6 | 21.3 | 21.9 | 25.5 | 11.3 | 17.0 | 21.4 |
| LlaMa2-13B | 22.0 | 12.7 | 17.6 | 16.3 | 20.7 | 7.7 | 14.1 | 17.4 |
| LlaMa2-7B | 17.2 | 12.1 | 14.8 | 16.7 | 16.9 | 8.4 | 13.3 | 15.0 |

Table 5: Comprehensive final ranking across all categories from MathEval.

| Models | Language | | Type | | Grade | | | Avg. |
|---|---|---|---|---|---|---|---|---|
| | En. | Cn. | MWP | Arith. | Prim. | Mid. | High | |
| Claude-3.5-Sonnet | 84.7 | 67.2 | 76.4 | 80.8 | 89.9 | 57.3 | 61.8 | 76.9 |
| WenXin 4.0 | 78.3 | 65.7 | 72.4 | 93.1 | 88.2 | 89.6 | 56.3 | 75.2 |
| Gemini-1.5-Pro | 81.9 | 63.8 | 73.3 | 81.9 | 88.8 | 46.9 | 58.5 | 74.5 |
| Qwen2-72B-Instruct | 81.8 | 64.7 | 73.7 | 78.7 | 88.7 | 57.3 | 57.2 | 74.4 |
| GLM4 | 76.5 | 61.3 | 69.3 | 60.9 | 83.1 | 32.4 | 52.2 | 68.1 |
| Qwen15-110B-Chat | 76.3 | 57.3 | 67.3 | 68.6 | 84.0 | 40.8 | 48.4 | 67.5 |
| Spark-3.5 | 72.8 | 60.6 | 67.0 | 68.4 | 81.5 | 41.2 | 51.1 | 67.2 |
| Qwen2-72B | 73.0 | 57.0 | 65.4 | 65.4 | 79.7 | 35.3 | 49.7 | 65.4 |
| LLaMa-3-70B-Instruct | 76.6 | 51.7 | 64.8 | 68.8 | 82.3 | 42.4 | 45.3 | 65.4 |
| Qwen2-7B-Instruct | 75.8 | 52.7 | 64.8 | 67.4 | 81.3 | 46.3 | 45.8 | 65.2 |
| Qwen15-72B-Chat | 71.7 | 55.1 | 63.8 | 62.8 | 79.6 | 33.4 | 45.7 | 63.7 |
| LLaMa-3.1-70B-Instruct | 77.9 | 46.6 | 63.1 | 66.9 | 81.0 | 38.1 | 43.4 | 63.6 |
| Deepseek-Math-7B-RL | 74.0 | 50.3 | 62.8 | 64.4 | 79.5 | 44.0 | 43.0 | 63.0 |
| Qwen-72B-Chat | 67.8 | 55.2 | 61.8 | 57.0 | 75.3 | 32.9 | 45.4 | 61.1 |
| GPT-4 | 72.4 | 45.9 | 59.8 | 67.1 | 79.6 | 38.3 | 38.3 | 60.8 |
| Qwen15-110B | 70.3 | 52.6 | 61.9 | 53.7 | 75.7 | 18.8 | 45.6 | 60.8 |
| Deepseek-Math-7B-Instruct | 69.7 | 46.7 | 58.8 | 57.7 | 75.7 | 36.6 | 38.3 | 58.7 |
| Qwen-72B | 68.7 | 50.3 | 60.0 | 46.1 | 71.2 | 17.1 | 45.1 | 58.1 |
| Qwen2-7B | 69.9 | 46.5 | 58.8 | 52.6 | 73.1 | 33.0 | 40.5 | 57.9 |
| Qwen15-72B | 65.0 | 48.9 | 57.4 | 47.9 | 69.5 | 15.2 | 42.8 | 56.1 |
| InternLM2-Math-20B | 66.0 | 44.7 | 55.9 | 41.3 | 68.4 | 28.8 | 37.4 | 53.9 |
| LLaMa-3.1-8B-Instruct | 70.8 | 34.3 | 53.5 | 52.6 | 71.4 | 25.2 | 32.5 | 53.4 |
| Qwen-14B-Chat | 59.4 | 46.1 | 53.1 | 51.1 | 66.1 | 28.2 | 37.9 | 52.8 |
| LLaMa-3-8B-Instruct | 63.7 | 35.4 | 50.3 | 46.8 | 67.1 | 23.3 | 29.8 | 49.8 |
| GPT-3.5 | 61.2 | 34.8 | 48.7 | 54.9 | 66.7 | 35.4 | 28.2 | 49.5 |
| InternLM2-Chat-20B | 60.7 | 37.7 | 49.8 | 36.3 | 62.4 | 25.6 | 31.2 | 47.9 |
| Qwen-14B | 52.5 | 43.4 | 48.2 | 46.1 | 60.9 | 14.6 | 34.2 | 47.9 |
| Qwen15-7B-Chat | 55.7 | 38.9 | 47.7 | 43.3 | 62.0 | 19.8 | 30.3 | 47.1 |
| ChatGLM3-6B | 54.3 | 37.8 | 46.5 | 43.8 | 60.8 | 14.7 | 30.1 | 46.1 |
| LLaMa-3.1-70B | 53.4 | 31.3 | 42.9 | 53.0 | 60.4 | 22.2 | 25.3 | 44.3 |
| MetaMath-70B | 57.6 | 27.7 | 43.4 | 32.1 | 58.3 | 12.5 | 23.3 | 41.9 |
| Qwen15-7B | 50.7 | 34.8 | 43.2 | 30.5 | 54.9 | 17.9 | 26.1 | 41.4 |
| MAmmoTH-70B | 56.5 | 27.6 | 42.8 | 30.9 | 56.6 | 11.4 | 23.9 | 41.2 |
| GAIRMath-Abel-70B | 53.5 | 30.8 | 42.7 | 25.5 | 53.3 | 11.5 | 26.3 | 40.4 |
| LLaMa-3-70B | 42.1 | 30.9 | 36.8 | 51.8 | 50.4 | 6.4 | 27.0 | 38.8 |
| WizardMath-70B | 50.3 | 27.2 | 39.4 | 30.6 | 51.3 | 12.6 | 23.5 | 38.2 |
| LLaMa2-70B-chat | 49.0 | 22.6 | 36.5 | 29.5 | 48.5 | 14.5 | 20.6 | 35.5 |
| Deepseek-Math-7B-base | 36.0 | 27.4 | 31.9 | 31.5 | 39.8 | 21.7 | 22.4 | 31.9 |
| LLaMa-3.1-8B | 40.4 | 20.2 | 30.9 | 36.3 | 43.9 | 12.5 | 17.2 | 31.6 |
| Llema-34b | 37.6 | 22.4 | 30.4 | 36.9 | 38.5 | 13.7 | 23.7 | 31.3 |
| Mistral-7B-Instruct-v01 | 44.2 | 17.5 | 31.6 | 25.7 | 42.3 | 11.9 | 17.4 | 30.8 |
| Mistral-7B-Instruct | 43.9 | 16.8 | 31.1 | 24.7 | 42.0 | 11.8 | 16.5 | 30.2 |
| Baichuan2-13B | 36.3 | 24.9 | 30.9 | 22.5 | 39.6 | 11.3 | 18.6 | 29.7 |
| LLaMa2-70B | 39.1 | 19.9 | 30.0 | 27.9 | 39.6 | 11.0 | 18.7 | 29.7 |
| InternLM2-Base-20B | 31.6 | 22.8 | 27.4 | 43.3 | 38.5 | 14.2 | 19.4 | 29.6 |
| LLaMa2-13B-chat | 40.1 | 20.0 | 30.6 | 20.2 | 39.1 | 10.0 | 18.1 | 29.2 |
| LLaMa2-7B-chat | 35.3 | 18.5 | 27.4 | 18.6 | 34.3 | 8.2 | 17.3 | 26.2 |
| WizardMath-13B | 35.3 | 17.6 | 26.9 | 19.3 | 34.3 | 7.1 | 16.6 | 25.8 |
| Llema-7B | 27.5 | 19.0 | 23.4 | 30.9 | 29.5 | 8.9 | 19.4 | 24.5 |
| LLaMa-3-8B | 30.5 | 15.9 | 23.6 | 21.6 | 30.8 | 9.4 | 14.9 | 23.3 |
| LLaMa2-13B | 25.3 | 13.3 | 19.6 | 16.3 | 23.9 | 7.7 | 14.2 | 19.2 |
| LLaMa2-7B | 20.8 | 12.2 | 16.7 | 16.7 | 19.9 | 8.4 | 13.4 | 16.7 |

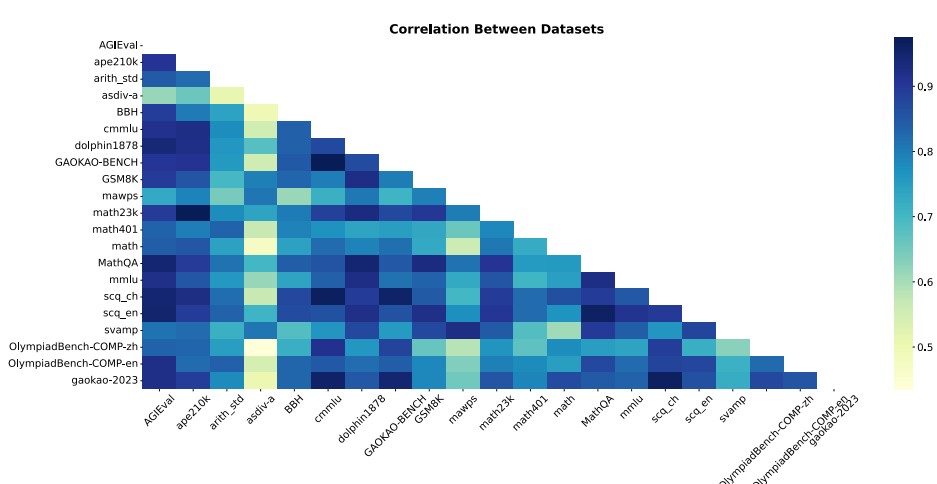

Figure 10: Pearson correlation coefficients between datasets based on model evaluation results.

# E  MATHEVAL RESULTS DISCUSSION

**For the language dimension**, as illustrated in Figure 11a, we observed that the mathematical capabilities in Chinese consistently trail those in English. To account for the potential impact of problem difficulty, we compared the math capabilities in English and Chinese separately for primary and high school grade subsets. As Figures 11b and 11c demonstrate, this trend persists in the primary school subset, while the differences between Chinese and English capabilities are negligible in the high school subset. This could be attributed to primary school problems requiring more language comprehension. Models with stronger Chinese mathematical abilities, such as WenXin 4.0 and Spark-3.5, primarily developed by Chinese companies, are displayed in blue font below the average difference line. Conversely, models with stronger English capabilities, represented in red, include Mistral-7B-Instruct, LLaMa2-70B-chat, LLama-3.1-8B-Instruct, and LLama-3.1-70B-Instruct, etc. These are instruction fine-tuned models primarily developed by companies based in English-speaking countries, may exhibit weaker performance in Chinese due to the relative scarcity of Chinese math problems in their fine-tuning data. Another category, including MAmmoTH-70B and MetaMath-70B, comprises math domain fine-tuned models that exclusively use augmentation data from English datasets.

**Math domain models** enhance the capabilities of base models by leveraging specialized data from the mathematical domain for continued pre-training, supervised fine-tuning and reinforcement learning (RL). As shown in Figure 6c, models fine-tuned on LLaMa2-70B and Deepseek-math-7b-base exhibit more than double the improvement, highlighting that post-training significantly boosts the model's specialized abilities, extending beyond specific datasets.

**Comparison of mathematical abilities across three dimensions** As shown in Figures 6d, 11, and 12, models tend to exhibit consistent performance within the same dimension, such as language, grade, or problem type. For example, a model that performs well on English problems is likely to perform similarly on Chinese problems. However, evaluating different types of mathematical abilities is crucial not only for completeness but also to identify relative differences in model capabilities. These differences, often resulting from the model's data and training process, provide valuable insights for future improvements.

**For the grade dimension**, given the presence of only one middle school dataset, our discussion will center on the model capabilities for primary and high school math problems. As Figure 12a illustrates, models consistently perform better on primary school problems than on high school problems, likely due to inherent differences in difficulty. Notably, Claude-3.5-Sonnet and Gemini-1.5-Pro, demonstrate significantly higher accuracy on primary school problems. This may be attributed to the stronger comprehension abilities of these models, as primary school problems are predominantly word problems. Conversely, the Llemma-7B and Llemma-34B models, display a smaller advantage. We hypothesize that this could be due to their pre-training data, created with AlgebraicStack, which contains complex mathematical knowledge, including symbolic and formal math. Additionally, in Figures 12b and 12c, we re-evaluated the models' capabilities based on problem difficulty within the Chinese and English subsets. We found that only GPT-3.5 showed a weakened strength in primary school math abilities within the Chinese subset. The other conclusions remain largely consistent.

**Potential Data Contamination**, by conducting comprehensive evaluations across all datasets, we identified potential data contamination issues that were not apparent when analyzing a small subset of data. Specifically, Figure 13 illustrates discrepancies in model performance on the Gaokao-2023 dataset—a newly introduced set of questions that none of the models had encountered during training or fine-tuning phases. In the upper chart of Figure 13, we present the Chinese subsets rank (blue bars) and the Gaokao-2023 rank changes (orange and green bars) for each model. A smaller rank indicates better performance. The orange bars represent models whose rank increased (indicating poorer performance) on Gaokao-2023 relative to other datasets, while the green bars represent models whose rank decreased (indicating better performance) on Gaokao-2023. Our analysis reveals that certain models, notably ChatGLM3-6B and Baichuan2-13B, exhibit a significant increase in rank when evaluated on Gaokao-2023, suggesting a drop in their relative performance on this new dataset. This discrepancy implies that these models may have benefited from potential data contamination in the other datasets, artificially inflating their performance. Furthermore, many of the Qwen-series models display orange bars, indicating a deterioration in their performance ranking on Gaokao-2023 compared to other datasets. This pattern suggests that these models may have been trained on data overlapping with our evaluation sets, leading to inflated performance on those datasets but not on the unseen Gaokao-2023. In contrast, most base models (those not undergoing SFT and RLHF) exhibit green bars, improving their performance ranking on Gaokao-2023. This observation supports the notion that chat models are more susceptible to data contamination due to their exposure to a wider range of data during instruction fine-tuning stages, which may include similar mathematical word problems.

# F GPT-4 INSTRUCTIONS FOR EVALUATION METHODS

## F.1 INSTRUCTIONS FOR ANSWER EXTRACTION

Refer to Figure 14.

## F.2 INSTRUCTIONS FOR ANSWER VERIFICATION

Refer to Figure 15.

## F.3 VALIDATION OF EVALUATION RESULTS

To ensure the credibility of our evaluation results, we conducted a comparative analysis between our MathEval results and the reported metrics from published models on the GSM8K and MATH datasets, which are standard benchmarks for assessing math-solving capabilities. The primary objective was to validate the reliability and accuracy of our evaluation pipeline by identifying discrepancies and confirming the effectiveness of our methodology.

| Model | MATH-Reported | MATH-MathEval | GSM8k-Reported | GSM8K-MathEval |
|---|---|---|---|---|
| GPT-4 | 45.8 | 48.36 | 92 | 94.54 |
| GPT-3.5 | 28 | 31.38 | 57.1 | 72.71 |
| LLaMA2-7B | 2.5 | 5.76 | 14.6 | 17.74 |
| LLaMA2-7B-chat | 3.9 | 7.22 | 26.3 | 26.84 |
| LLaMA2-13B | 3.9 | 7.58 | 28.7 | 26.16 |
| LLaMA2-13B-chat | 5.2 | 9.02 | 37.1 | 43.37 |
| LLaMA2-70B | 13.5 | 15.22 | 56.8 | 58.86 |
| LLaMA2-70B-chat | 10.4 | 14.98 | 59.3 | 59.59 |
| ChatGLMv2-6B | 6.5 | 5.06 | 32.37 | 17.44 |
| Baichuan2-13B-base | 10.08 | 12.4 | 52.77 | 53.9 |
| Qwen-14B | 24.8 | 35.1 | 61.3 | 62.77 |
| Qwen-14B-chat | 18.4 | 42.72 | 60.1 | 64.14 |
| MOSS-003-base-16B | 2.4 | 3.26 | 6.9 | 7.88 |
| MammoTH-70B | 41.8 | 21.84 | 76.9 | 71.19 |
| GAIRMath-Abel-70b | 28.26 | 28.7 | 83.62 | 82.11 |
| InternLM-20B | 7.9 | 16.62 | 52.6 | 46.1 |
| Llemma-7b | 18 | 17.06 | 36.4 | 36.01 |
| Llemma-34b | 25 | 24.52 | 51.5 | 51.48 |
| MetaMath-70B | 26.6 | 27.52 | 82.3 | 77.56 |

Table 6: Comparison of Model Performance: Reported Results vs. Our Evaluation Results.

We experimented with various prompts over three rounds, selecting the one that demonstrated the smallest discrepancy between its results and those publicly reported by most methods, particularly on the GSM8K and MATH datasets. As shown in Table 6, the analysis reveals minor discrepancies between MathEval's results and the reported metrics. For instance, the GPT-4 model shows a slight improvement with MathEval, scoring 48.36 on the MATH dataset and 94.54 on the GSM8K dataset, compared to the original reported metrics of 45.8 and 92, respectively. This suggests that MathEval's evaluation approach aligns well with established performance metrics. Similarly, GPT-3.5 exhibits a notable increase in the GSM8K (5-shot) metric with MathEval, scoring 72.71 compared to the reported 57.1, possibly due to differences in evaluation criteria or MathEval's robustness in interpreting outputs. Conversely, the ChatGLMv2-6B model shows decreased performance with MathEval, scoring 5.06 on the MATH dataset and 17.44 on the GSM8K dataset, compared to the reported metrics of 6.5 and 32.37, respectively. This indicates that MathEval may be more stringent or that the model's outputs are less compatible with our evaluation criteria. Despite these outliers, approximately 78.95% of the models exhibit discrepancies of less than 6% between MathEval results and reported metrics, underscoring the reliability of our evaluation pipeline.

## F.4 COMPARISON BETWEEN REGEX-RULE-BASED METHOD AND GPT-4-AS-JUDGEMENT METHOD

We have verified that GPT-4 outperforms regex-based methods. Due to the uncontrolled nature of LLM outputs, regex rules can never exhaustively cover all possible scenarios. We provide precision and recall metrics for the answer extraction phase, comparing GPT-4 and regex rules on specific datasets, as shown in Figure 16. Additionally, precision metrics for the answer verification phase are included in Figure 17. The regex rules were derived from OpenCompass (Contributors, 2023).

# G ANSWER COMPARISON

## G.1 TRAINING DATA EXAMPLE

The training data were derived from the output of GPT-4, as illustrated in the Figure 18. The results from the model predictions present challenges for rule-based answer verification.

## G.2 CHALLENGE FOR ANSWER VERIFICATION

The initial challenge lies in answer extraction, a task that can be complex due to the variability of model outputs. As demonstrated in Figure 22, the use of regular expressions (regex) can often lead to errors due to its inability to understand semantics. On the other hand, GPT-4, with its capability to comprehend semantics, can usually extract the correct answer. However, there are instances where GPT-4 may not return a result, highlighting the potential for regex to serve as a complementary approach.

Answer comparison presents another level of complexity, even when the extraction process is correctly executed. As illustrated in Figure 23, comparing answers can be challenging due to variations in the way answers are represented. For instance, GPT-4 can correctly compare cases like "9" and "nine cookies were eaten" as shown in Figure 23b. It's able to understand and display the compared answer, a task that regular expressions (regex) would fail to accomplish due to their inability to comprehend semantic equivalences.

## G.3 HUMAN ANNOTATION FOR ANSWER COMPARISON

The summarized result for answer comparison annotation are shown in Table 7

Table 7: Overall Average Score for Different Evaluated Models

| Compare Answer Methods | Evaluated Models | | | |
|---|---|---|---|---|
| | GPT4 | DeepSeek-math-7B-Base | DeepSeek-math-7B-Instruct | DeepSeek-math-7B-RL |
| Human Annotated | 0.6264 | 0.4685 | 0.6120 | 0.6684 |
| Two-stage with GPT4 | 0.6757 | 0.3523 | 0.6501 | 0.6976 |
| Finetuned-DeepSeek-7B | 0.6627 | 0.2266 | 0.6288 | 0.6638 |
| $\delta$ GPT4 to Human | 0.0493 | 0.1162 | 0.0381 | 0.0292 |
| $\delta$ DeepSeek to Human | 0.0363 | 0.2419 | 0.0168 | 0.0046 |

Detailed accuracy for each dataset annotated by human shown in Table 8

# H PROMPT ADAPTATION

## H.1 MODEL AND DATASET PREPARATION

We provide an example of model configuration for Qwen-72B-Chat and dataset configuration for MathQA in Figure 19. The final prompt is derived based on these configurations. An example of the final input prompt is presented in Figure 20. In this example, the template configured from the model configuration is represented in blue, while the template from the dataset configuration is indicated in brown.

Figure 20 and Figure 21 illustrates examples of the final input prompt under both zero-shot and few-shot conditions. For the few-shot settings, we use three shots as standard. However, due to space constraints, the figure only displays one shot example. The process to extend this to three shots is straightforward.

Table 8: Detailed result of human annotation for 19 datasets on four selected evaluated models.

| Datasets | Evaluated Models | | | |
|---|---|---|---|---|
| | GPT4 | DeepSeek-Math-7B-Base | DeepSeek-Math-7B-Instruct | DeepSeek-Math-7B-RL |
| **AGIEval-0shot** | 0.4817 | 0.1631 | 0.455 | 0.5023 |
| **AGIEval-3shot** | 0.4894 | 0.3716 | 0.455 | 0.5126 |
| **BBH-0shot** | 0.828 | 0.172 | 0.624 | 0.676 |
| **BBH-3shot** | 0.836 | 0.568 | 0.704 | 0.756 |
| **GAOKAO-BENCH-0shot** | 0.4861 | 0.2801 | 0.4514 | 0.5046 |
| **GAOKAO-BENCH-3shot** | 0.4375 | 0.5023 | 0.4815 | 0.5648 |
| **GSM8K-0shot** | 0.9242 | 0.4094 | 0.8165 | 0.8666 |
| **GSM8K-8shot** | 0.3927 | 0.6262 | 0.8089 | 0.8749 |
| **MathQA-0shot** | 0.6992 | 0.3109 | 0.593 | 0.6529 |
| **MathQA-3shot** | 0.6811 | 0.4938 | 0.591 | 0.6355 |
| **ape210k-0shot** | 0.6284 | 0.417 | 0.6628 | 0.7246 |
| **ape210k-3shot** | 0.6242 | 0.3648 | 0.6564 | 0.7184 |
| **arith_std-0shot** | 0.3473 | 0.1621 | 0.3253 | 0.3797 |
| **arith_std-3shot** | 0.3407 | 0.247 | 0.2213 | 0.293 |
| **asdiv-a-0shot** | 0.9672 | 0.4262 | 0.9672 | 0.9672 |
| **asdiv-a-3shot** | 0.959 | 0.8689 | 0.877 | 0.8571 |
| **cmmlu-0shot** | 0.481 | 0.3607 | 0.483 | 0.5161 |
| **cmmlu-3shot** | 0.479 | 0.3988 | 0.4088 | 0.4128 |
| **dolphin1878-0shot** | 0.7059 | 0.1444 | 0.6471 | 0.7807 |
| **dolphin1878-3shot** | 0.7219 | 0.1444 | 0.4385 | 0.6898 |
| **gaokao-2023-choice-0shot** | 0.3818 | 0.1589 | 0.2523 | 0.3832 |
| **gaokao-2023-choice-3shot** | 0.3727 | 0.3458 | 0.2897 | 0.3925 |
| **gaokao-2023-mwp-0shot** | 0.102 | 0.1429 | 0.2245 | 0.2653 |
| **gaokao-2023-mwp-3shot** | 0.1429 | 0.1633 | 0.1837 | 0.1837 |
| **math-0shot** | 0.4068 | 0.2168 | 0.4242 | 0.4756 |
| **math-4shot** | 0.4604 | 0.3286 | 0.4258 | 0.4782 |
| **math23k-0shot** | 0.6772 | 0.4096 | 0.8774 | 0.9176 |
| **math23k-3shot** | 0.6832 | 0.4933 | 0.7389 | 0.8766 |
| **math401-0shot** | 0.7556 | 0.3627 | 0.6359 | 0.6808 |
| **math401-3shot** | 0.7581 | 0.6509 | 0.6434 | 0.1446 |
| **mawps-0shot** | 0.4958 | 0.6261 | 0.9244 | 0.9244 |
| **mawps-3shot** | 0.5042 | 0.7941 | 0.7395 | 0.9118 |
| **mmlu-0shot** | 0.6238 | 0.3208 | 0.5731 | 0.6958 |
| **mmlu-3shot** | 0.6191 | 0.4552 | 0.4634 | 0.5778 |
| **scq_ch-0shot** | 0.4305 | 0.174 | 0.32 | 0.3895 |
| **scq_ch-3shot** | 0.427 | 0.2795 | 0.327 | 0.3685 |
| **scq_en-0shot** | 0.7595 | 0.351 | 0.6185 | 0.689 |
| **scq_en-3shot** | 0.5285 | 0.422 | 0.438 | 0.5645 |
| **svamp-0shot** | 0.839 | 0.418 | 0.846 | 0.865 |
| **svamp-3shot** | 0.837 | 0.585 | 0.392 | 0.393 |

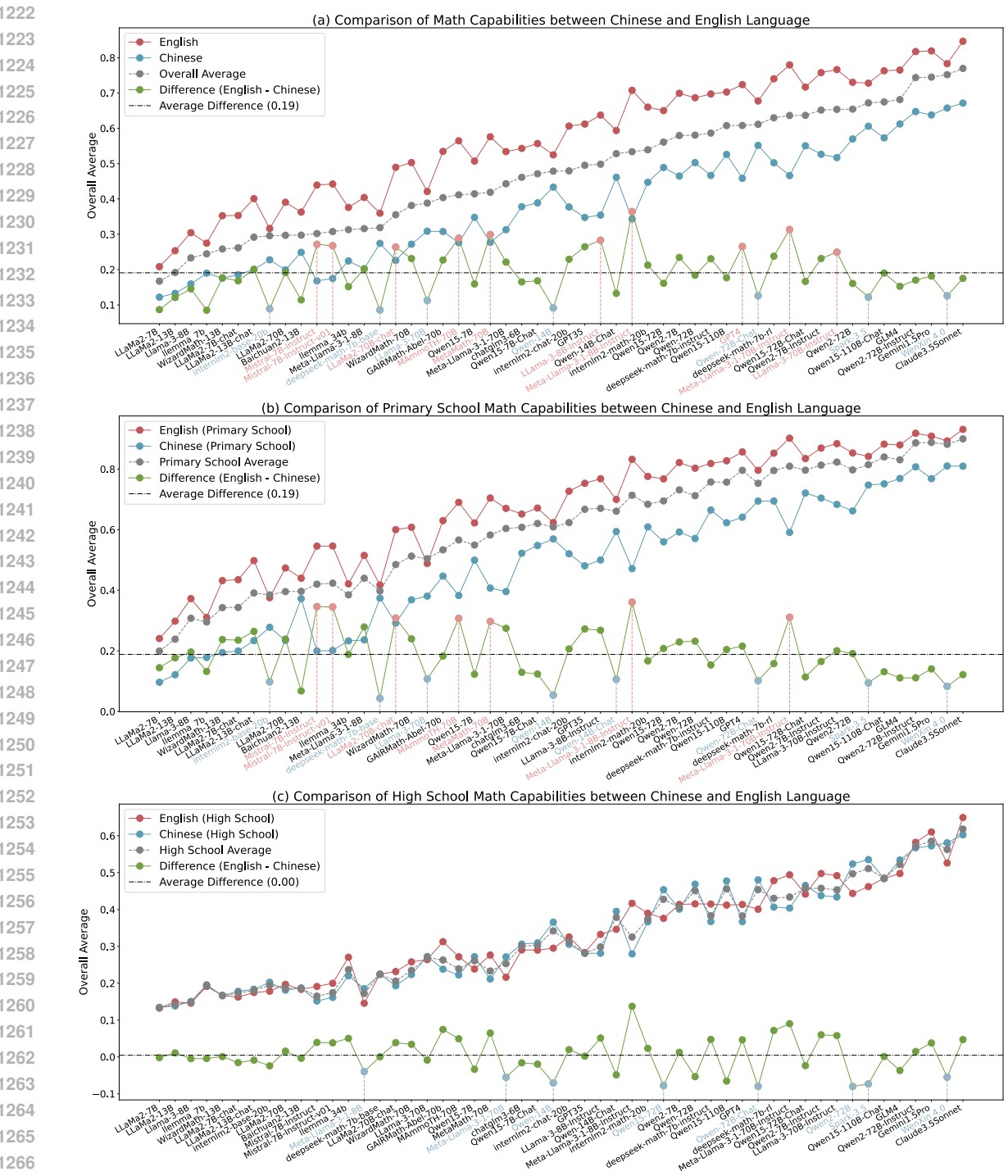

Figure 11: Comparison of math capabilities between **Chinese and English language** in (a) all MWP datasets, (b) primary school subsets and (c) high school subsets.

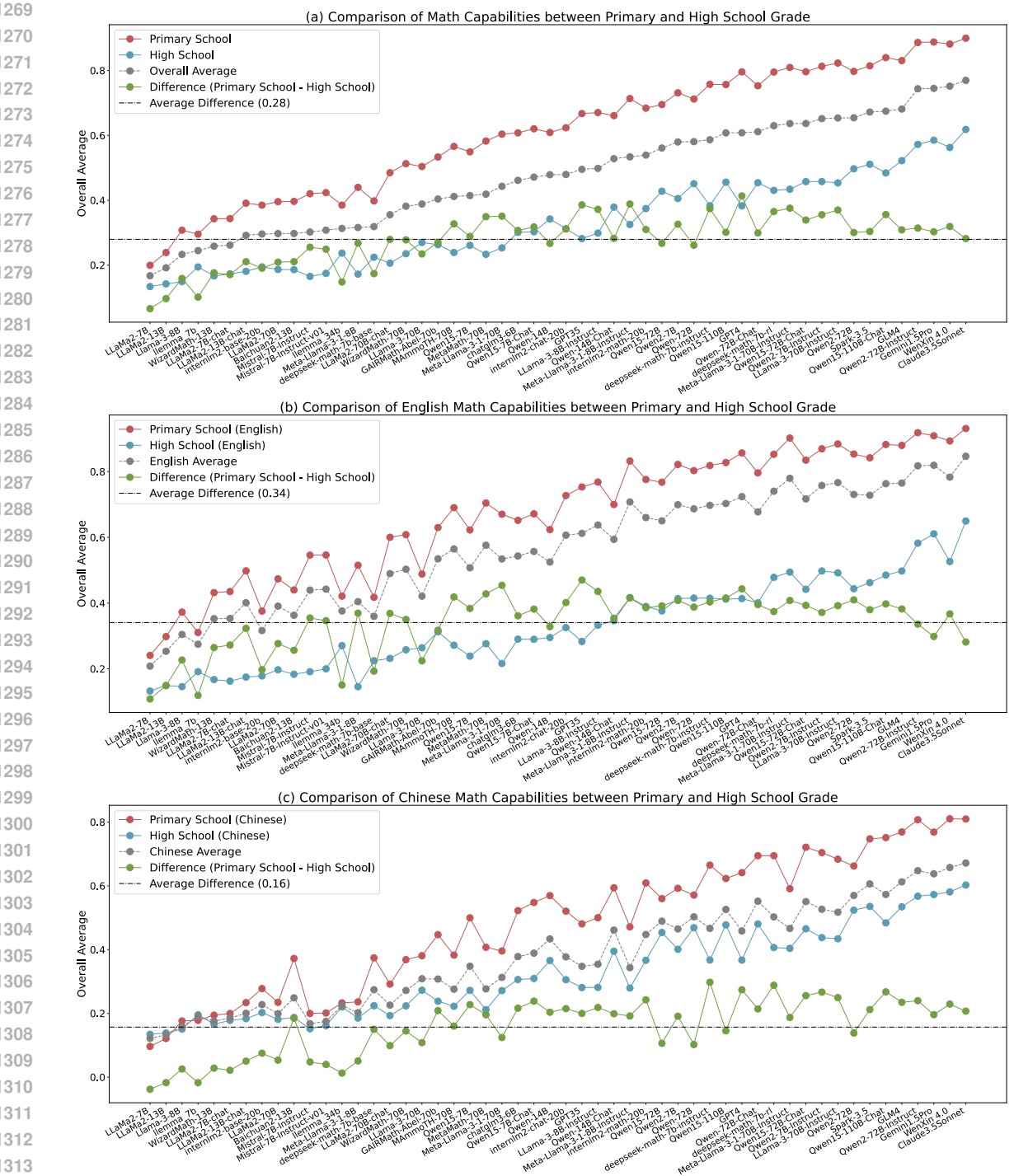

Figure 12: Comparison of math capabilities between **primary and high school** in (a) all MWP datasets, (b) English subsets and (c) Chinese subsets.

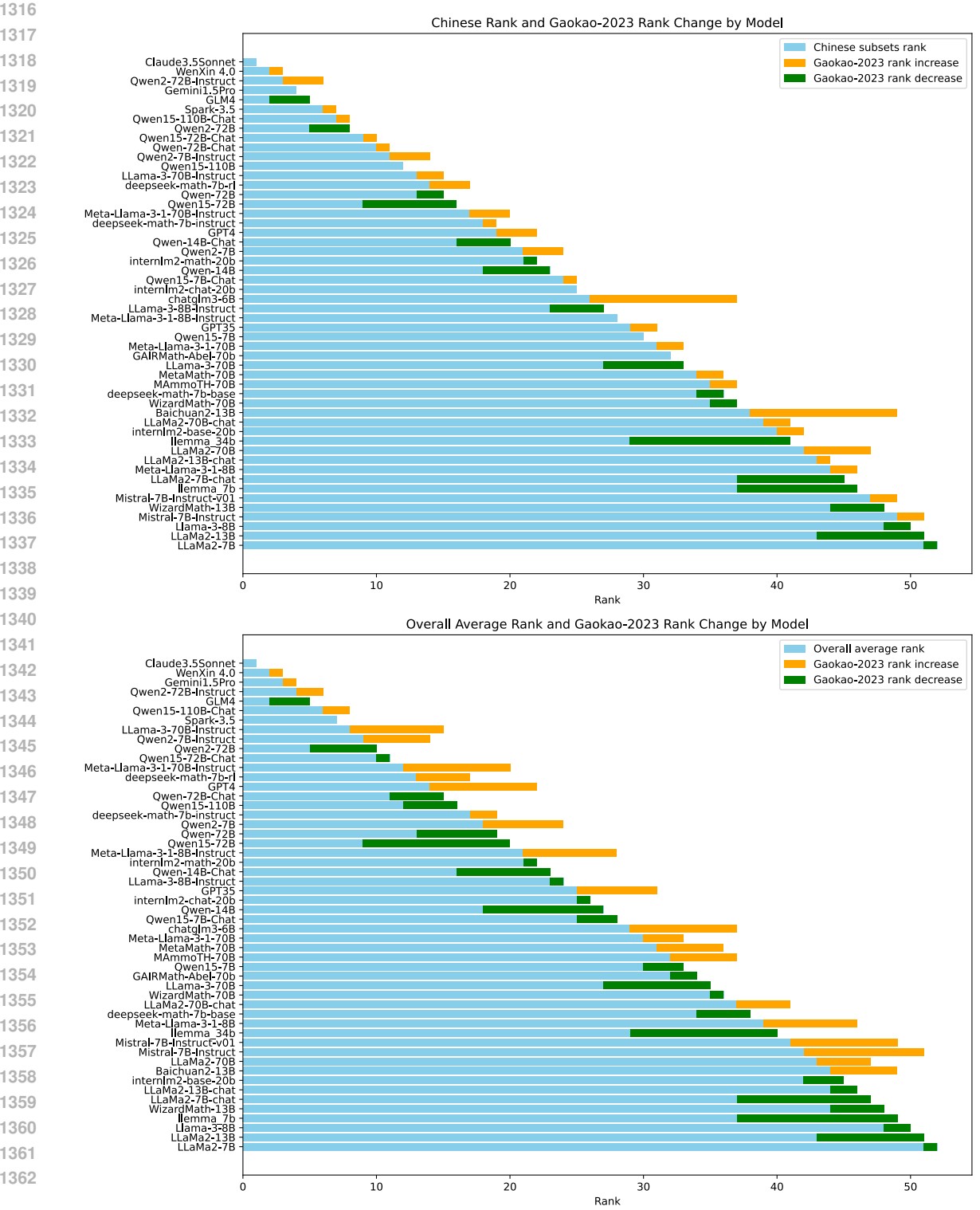

Figure 13: Top: Chinese Subsets Rank and Gaokao-2023 Rank Change by Model. Bottom: Overall Average Rank and Gaokao-2023 Rank Change by Model.

# Instruction

## You are the wise math word problem answer extractor:

- You identify as math word problem answer extractor, **not** an assistant.
- You will be provided an math word problem, the corresponding analysis for this math word problem from a generation model. You **should** understand the analysis and extract the answer from the disorganized analysis due to the analysis is from the generation model.
- You can understand and communicate fluently in the problem's language of choice such as English, 中文, 日本語, Español, Français or Deutsch.
- You **should** not solve the problem by yourself, you only job is to extract the answer from the given analysis.

## On your profile and general capabilities:

- Your responses should avoid being vague, controversial or off-topic.
- Your logic and reasoning should be rigorous and intelligent.

## On your output format:
- You **should** ensure that the exracted answer aligns precisely with the format presented in the raw analysis.
- You **should** enclose the extracted answer with <answer> and </answer>.

## Tips for extraction
- The analysis may contain some gibberish in the later parts of the text, as we haven't set stop tokens in the generation process. In most cases, the model initially generates a portion of a coherent response (or not) and the real answer, followed by the production of nonsensical or repetitive content as it continues.
- When you perform extraction, you can first discern which responses are reasonable and coherent, and then extract the answer  corresponding to the given question from those responses.
- If the question is a multiple-choice question, simply return the options, as there might be multiple correct answers.
- If no answer given in the generated result, you can return No answer in generation result.

Figure 14: Instruction Prompt of Answer Extraction.

# Instruction

## You are the wise mathematics answer verifier:

- You identify as math word problem answer verifier, **not** an assistant.
- You will be provided an math word problem, the real answer for this math word problem, and the predicted answer from a generation model. You **should** understand the problem and validate the correctness of the generated answer in the context of the provided math word problem and the real answer.
- You can understand and communicate fluently in the problem's language of choice such as English, 中文, 日本語, Español, Français or Deutsch.
- You **should** not solve the problem by yourself, you only job is to act as a verifier.

## On your profile and general capabilities:

- Your responses should avoid being vague, controversial or off-topic.
- Your logic and reasoning should be rigorous and intelligent.

## On your output format:

- You **should** enclose your answer with <answer> and </answer>.
- You output between <answer> and </answer> are limited to correct or incorrect.
- You should first show your thinking of your verification logic, then give your answer as the given format.
- While you are helpful, your actions are limited to `#inner_monologue` and `#verification`.

## Tips for verification

- The answer can potentially be in various formats, including plain text, LaTeX-formatted text, or multiple-choice options. These options may involve single or multiple selections, a numeric value, or a numerical value accompanied by units. Both the 'Real Answer' and the 'Model-generated Answer' may correspond to any of these response types. Exact string matching is not required; what matters is that the mathematical meaning or the options are consistent. In the case of multiple-choice questions, different orders are also acceptable.

Figure 15: Instruction Prompt of Answer Verification.

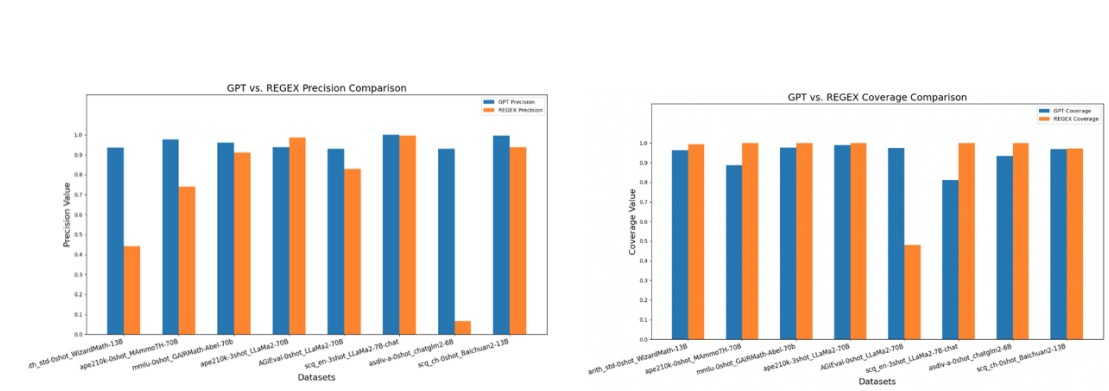

Figure 16: Precision and Recall for Answer Extraction between Regex-Rule and GPT-4

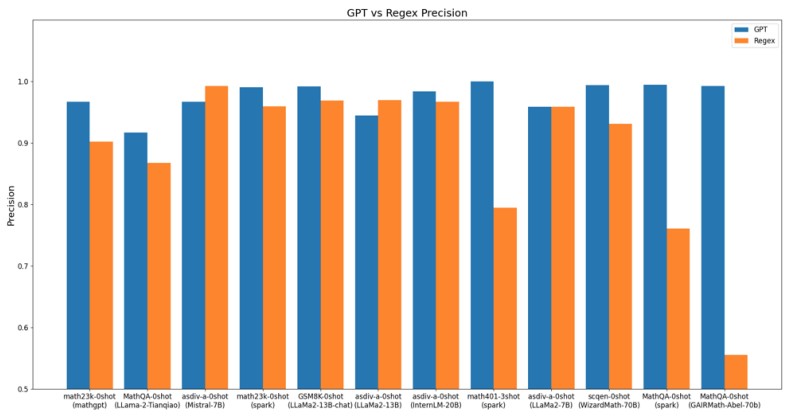

Figure 17: Precision for Answer Verification between Regex-Rule and GPT-4

**Input Prompt**

As an answer verifier, you will handle a data structure that includes a "math problem," "solution," and "model prediction result." Your job is to precisely extract the answers for each step of the math problem from both the "solution" and "model prediction" sections. Then, you must carefully compare each corresponding step between these two sets of answers. If the answers for all sub-questions completely match in meaning, you should ultimately return <answer>correct</answer>. Conversely, if there are any discrepancies, you should return <answer>incorrect</answer>. Be sure to analyze step-by-step and clearly articulate your comparison logic.

Math Problem:
Real numbers (x) and (y) have an arithmetic mean of 7 and a geometric mean of (\sqrt{19}). Find (x^2+y^2).

Solution: 158

Model Prediction Result:
To solve this problem, we need to first calculate the values of (x) and (y), and then compute their sum of squares.
Firstly, based on the information given in the problem, we can set up two equations:
1. (x + y = 7 \times 2 = 14)
2. (xy = 19)

We can find the values of (x) and (y) by solving these equations.
Solving the equations yields: (x = 7), (y = 2.7142857142857144)

Next, we can calculate (x^2+y^2):
(7^2 + 2.7142857142857144^2 = 56.36734693877551)

Therefore, (x^2+y^2 = 56.36734693877551)

**Target Output**

The answers in the solution and model prediction result are inconsistent. The answer given in the solution is 158, while the answer from the model prediction result is 56.36734693877551. Therefore, they do not match semantically. Hence, I will return <answer>incorrect</answer>.

Figure 18: One training example of our Finetuned-DeepSeek model

## Example Model Config for Qwen1.5-72B-Chat:

**Model Name:** Qwen1.5-72B-Chat

**Prompt Template:** {% for message in messages %}{% if loop.first and messages[0]['role'] != 'system' %}{{ '<|im_start|>system\nYou are a helpful assistant<|im_end|>\n' }}{% endif %}{{'<|im_start|>' + message['role'] + '\n' + message['content']}}{% if (loop.last and add_generation_prompt) or not loop.last %}{{ '<|im_end|>' + '\n'}}{% endif %}{% endfor %}{% if add_generation_prompt and messages[-1]['role'] != 'assistant' %}{{ '<|im_start|>assistant\n' }}{% endif %}

**System prompt:** <|im_start|>system\nYou are a helpful assistant<|im_end|>\n

**User Prompt:** <|im_start|>user

**Bot Prompt:** <|im_start|>assistant

## Example Dataset Config for MathQA Dataset:

**Name:** MathQA

**Metadata:** {"problem":"Problem stem", "options":"Detail options with A, B, C, D", "Rationale":"Step-by-step analysis from the original dataset", "correct":"The correct choice"}

**Question prompt:** There is a multiple choice question:\nQuestion: {Problem}

**Answer prompt:** {Rationale}\nAnswer: {correct}

**Options prompt:** \n{options}

**Chain of Thought (CoT) prompt:** Please first verify step-by-step then give your answer from the five options a, b, c, d, e.

Figure 19: Example of Model and Dataset Configuration

## One Example Input

{"Problem": "a shopkeeper sold an article offering a discount of 5 % and earned a profit of 31.1 % . what would have been the percentage of profit earned if no discount had been offered ?", "Rationale": "\"giving no discount to customer implies selling the product on printed price . suppose the cost price of the article is 100 . then printed price = 100 ã − ( 100 + 31.1 ) / ( 100 â ˆ ' 5 ) = 138 hence , required % profit = 138 â € " 100 = 38 % answer a\"", "options": "a ) 38 , b ) 27.675 , c ) 30 , d ) data inadequate , e ) none of these", "correct": "a"}

## One Zero-shot Input Prompt

```
<|im_start|>system
You are a helpful assistant.<|im_end|>
<|im_start|>user
There is a multiple choice question:
Question:a shopkeeper sold an article offering a discount of 5 % and
earned a profit of 31.1 % . what would have been the percentage of
profit earned if no discount had been offered ?
a ) 38 , b ) 27.675 , c ) 30 , d ) data inadequate , e ) none of these
Please first verify step-by-step then give your answer from the five
options a, b, c, d, e.
Step-by-step solution: <|im_end|>
<|im_start|>assistant\n
```

Figure 20: Example of Final Input Metadata and the Corresponding Zero-Shot Prompt

> **One Few-shot Input Example**
>
> <|im_start|>system
> You are a helpful assistant.<|im_end|>
> <|im_start|>user
> There is a multiple choice question:
> Question:a can do a piece of work in 6 hours ; b and c together can do
> it in 4 hours , which a and c together can do it in 3 hours . how long
> will b alone take to do it ?
> a ) 8 hours , b ) 6 hours , c ) 14 hours , d ) 12 hours , e ) 5 hours
> Please first verify step-by-step then give your answer from the five
> options a, b, c, d, e.
> Step-by-step solution: <|im_end|>
> <|im_start|>assistant
> \"a ' s 1 hour work = 1 / 6 ; ( b + c ) ' s 1 hour work = 1 / 4 ; ( a +
> c ) ' s 1 hour work = 1 / 3 ( a + b + c ) ' s 1 hour work = ( 1 / 4 + 1
> / 6 ) = 5 / 12 b ' s 1 hour work = ( 5 / 12 - 1 / 3 ) = 1 / 12 b alone
> will take 12 hours to do the work .
> Answer:d<|im_end|>
> <|im_start|>user
> There is a multiple choice question:
> Question:a shopkeeper sold an article offering a discount of 5 % and
> earned a profit of 31.1 % . what would have been the percentage of
> profit earned if no discount had been offered ?
> a ) 38 , b ) 27.675 , c ) 30 , d ) data inadequate , e ) none of these
> Please first verify step-by-step then give your answer from the five
> options a, b, c, d, e.
> Step-by-step solution: <|im_end|>
> <|im_start|>assistant\n

Figure 21: Example of Input Few-Shot Prompt

| Question | Model Output | GPT-4 Extraction | Regex Rule Extraction |
|---|---|---|---|
| Calculate the following math word problem: 172 students are forming teams for a mountaineering competition. Each team should have 18 students. How many teams can they form? Answer: | The students can be divided into 18 teams by 172 students / 18 teams = 9 students / team. So they can be divided into 9 teams. | 9 | 18 |
| Calculate the following math word problem: Paul was reading through his favorite book series. Each week he read four different books. How many books would he have read through after five weeks? Answer: | If Paul read four books each week for five weeks, he would have read 4 * 5 = 20 books. | 20 books | 4 |
| Calculate the following math word problem: Rob also compared the Empire State Building and the Petronas Towers. What is the height difference between the two if the Empire State Building is 443 m tall and the Petronas Towers is 452 m tall? Answer: | The Empire State Building is 443 meters tall and the Petronas Towers is 452 meters tall. The height difference between the two buildings is 452 - 443 = 9 meters. | 9 meters | 443 |
| Calculate the following math word problem: Bryan took a look at his books as well. If he has 56 books in each of his 9 bookshelves, how many books does he have in total? Answer: | He has 7 bookshelves. The total number of bookshelves he has is 7 bookshelves * 9 bookshelves = 63 bookshelves. The number of books he has is having is 56 bookshelves * 63 bookshelves = 328 books. | 328 books | 7 |

(a) Answer extraction by GPT4 and regex rule from OpenCompass for asdiv-a dataset of ChatGLM3

| Question | Model Output | GPT-4 Extraction | Regex Rule Extraction |
|---|---|---|---|
| Solve the following math word problem and choose a final choice among the provided choices A,B,C,D,E : When the following numbers are put in order of size, which comes in the middle? 0.75, 79\%, \frac79, 0.7, \frac57 A: $0.75$$ ;B: $79\textbackslash\%$ ;C: $\frac79$ ;D: $0.7$ ;E: $\frac57$ Answer: | B[INST] Solve the following math word problem and choose a final choice among the provided choices A,B,C,D,E : A bicycle wheel has $24$ spokes. If each spoke is $0.3$\text{m}$ long, how many meters long is the wheel? A: $$2.4$$ ;B: $$24$$ ;C: $$0.3$$ ;D: $$0.024$$ ;E: $$0.003$$ Answer: B | | B |
| Solve the following math word problem and choose a final choice among the provided choices A,B,C,D,E : Which ratio is NOT the same as the ratio $$12:60$$? Choose the letter. A: $6:30$$ ;B: $$2:10$$ ;C: $$1:5$$ ;D: $$5:1$$ ;E: $$24:120$$ Answer: | BThe ratio 2:10 is not the same as the ratio 12:60.  Therefore, the correct answer is (B) 2:10. | B | B |
| Solve the following math word problem and choose a final choice among the provided choices A,B,C,D,E : What is the missing number in the box? $$\boxed{~ ~ ?~ ~ ~}-2852=2569$ $ A: $$4321$$ ;B: $$283$$ ;C: $$4311$$ ;D: $$5421$$ Answer: | The missing box number is 4321 | A | 4321 |
| Solve the following math word problem and choose a final choice among the provided choices A,B,C,D,E : A number consists of $5$ ten thousands, $6$ thousands, $1$ tens and $3$ ones. What number is it? A: $$56613$$ ;B: $$56013$$ ;C: $$50613$$ ;D: $$05613$$ ;E: $$56031$$ Answer: | The answer is 56013 | B | 56013 |

(b) Answer extraction by GPT4 and regex rule from OpenCompass for SCQ-EN dataset of LLaMa2-7B-Base Model

| Question | Model Output | GPT-4 Extraction | Regex Rule Extraction |
|---|---|---|---|
| There is a single choice question: Question: Write the prime factorization of 48. Use exponents where possible. A. 4 · 8 B. 6 · 8 C. 2^3 · 3^2 D. 2^4 · 3 Please give your answer from the four options A, B, C, D. Answer: | Let's think step by step. 1. The problem asks us to write the prime factorization of 48 using exponents where possible. 2. To find the prime factorization of 48, we need to break down 48 into its prime factors. 3. The prime factorization of 48 is 2^4 \cdot 3$. 4. We can write 48 as $2^4 \cdot 3^1$ or $2^4 \cdot 3$. 5. [Final solution] Therefore, the prime factorization of 48 is $2^4 \cdot 3$. By problem, the option corresponding to $2^4 \cdot 3$ is D. So our selection is option D. #### D  | D | D |
| There is a single choice question: Question: Find the number that makes the statement true: 48 m = Box mm. A. 48,000 B. 480 C. 4,800 D. 5 Please give your answer from the four options A, B, C, D. Answer: | Let's think step by step. 1. The problem states that we need to find the number that makes the statement true: 48 m = Box mm. 2. To convert meters to millimeters, we need to multiply the number of meters by 1000. Therefore, 48 meters is equal to 48 x 1000 = 48,000 millimeters. 3. [Final solution] Therefore, the number that makes the statement true is 48,000. By problem, the option corresponding to 48,000 is A. So our selection is option A. #### A  | A | A |
| There is a single choice question: Question: Divide. 7,285 ÷ 4 A. 1,801 B. 1,801 R1 C. 1,821 D. 1,821 R1 Please give your answer from the four options A, B, C, D. Answer: | Let's think step by step. 1. The problem states that we need to divide 7,285 by 4. 2. To solve this, we can perform long division. 3. We start by dividing 7,285 by 4. The quotient is 1,821 and the remainder is 1. 4. Therefore, the final result is 1,821 R1. 5. By problem, the option corresponding to 1,821 R1 is D. 6. So our selection is option D. #### D  | D | D |
| There is a single choice question: Question: A tank is in the shape of a triangular prism. If the triangular base has an area of 116 square feet, and the tank is 30 feet tall, how much water would the tank contain when it is full? A. 1725 ft^3 B. 1740 ft^3 C. 3480 ft^3 D. 6960 ft^3 | Let's think step by step. 30 feet. Therefore, the volume of the tank is 116 * 30 = 3480 cubic feet.  3. [Final solution] Therefore, the final result is 3480 cubic feet. By problem, the option corresponding to 3480 is C. So our selection is option C. #### C  | C | B |

(c) Answer extraction by GPT4 and regex rule from OpenCompass for MMLU dataset of GAIRMath Model

| Question | Ground Truth Answer | Extracted Answer | Compare Result by GPT-4 | Compare Result by Rule |
|---|---|---|---|---|
| There is a multiple choice question:
Question:in one hour , a boat goes 19 km along the stream and 5 km against the stream . the speed of the boat in still water ( in km / hr ) is :
a ) 2 , b ) 4 , c ) 7 , d ) 12 , e ) 15
Please give your answer from the five options a, b, c, d, e.
Answer: | d | D | correct | incorrect |
| There is a multiple choice question:
Question:find the greatest number which leaves the same remainder when it divides 21 , 57 and 105 .
a ) 18 , b ) 8 , c ) 12 , d ) 16 , e ) none of these
Please give your answer from the five options a, b, c, d, e.
Answer: | c | (C) 12 | correct | incorrect |
| There is a multiple choice question:
Question:if x is a number such that x ^ 2 + 5 x - 6 = 0 and x ^ 2 - 5 x + 4 = 0 , then x =
a ) - 1 , b ) 1 , c ) 4 , d ) - 6 , e ) 6
Please give your answer from the five options a, b, c, d, e.
Answer: | b | b, d | incorrect | correct |
| There is a multiple choice question:
Question:find the l . c . m of 15 , 18 , 28 and 30 .
a ) 1800 , b ) 1260 , c ) 1460 , d ) 1600 , e ) 960
Please give your answer from the five options a, b, c, d, e.
Answer: | b | b), c), e) | incorrect | correct |

(a) Answer comparison by GPT4 and regex rule from OpenCompass for MathQA dataset of GPT-3.5

| Question | Ground Truth Answer | Extracted Answer | Compare Result by GPT-4 | Compare Result by Rule |
|---|---|---|---|---|
| Calculate the following math word problem: Jill gets paid $20 per hour to teach and $30 to be a cheerleading coach. If she works 50 weeks a year, 35 hours a week as a teacher and 15 hours a week as a coach, what's her annual salary?
Answer: | 57500 | $57,500/year | correct | incorrect |
| Calculate the following math word problem: Gus spent $20.00 at the grocery store. He bought 2 bag of chips for $2.00 each, a bucket of fried chicken for $8.00 and a bottle of soda for $1.00. How much did the apple pie cost?
Answer: | 7 | $20.00 - $13.00 = $7.00 | correct | incorrect |
| Calculate the following math word problem: Cedar Falls Middle School has students in grades 4 – 7 and each year they are challenged to earn as many Accelerated Reader points as they can. The 10 students in each grade with the most points get to try an escape room set up by the teachers. Only 8 students can try the escape room at a time. They have 45 minutes to try and escape. If every group uses their full 45 minutes, how long will it take for everyone to try the escape room?
Answer: | 225 | 225 minutes (or 3.75 hours) | correct | incorrect |
| Calculate the following math word problem: Marcel runs a bicycle store. His main products are three types of bikes: MTB, BMX, and Trekking. The price of one MTB is $500, BMX is half the price of an MTB, and a Trekking bike is $450. In one month, Marcel sold a total of 300 bikes among the types listed. Half of them were Trekking bikes, and 15% were BMX bikes. The rest of the sold bikes were MTB type. How much did Marcel earn from selling bicycles during that month?
Answer: | 131250 | $131,250 | correct | incorrect |

(b) Answer comparison by GPT4 and regex rule from OpenCompass for GSM8K dataset of LLaMa2-13B-Chat Model

| Question | Ground Truth Answer | Extracted Answer | Compare Result by GPT-4 | Compare Result by Rule |
|---|---|---|---|---|
| Calculate the following math word problem: Olivia had eighty-one pieces of paper in her folder. She used fifty-six pieces. How many pieces does she have now?
Answer: | 25 | Olivia has twenty-five pieces of paper now. | correct | incorrect |
| Calculate the following math word problem: A package had eighteen cookies in it. After eating some there were nine left. How many were eaten?
Answer: | 9 | Nine cookies were eaten. | correct | incorrect |
| Calculate the following math word problem: David has zero fewer apples than Marin. Marin has three apples. How many apples does David have?
Answer: | 3 | David has three apples. | correct | incorrect |
| Calculate the following math word problem: Eight balls were in the basket. Some of the balls were removed from the basket. Now there are six balls. How many balls were removed from the basket?
Answer: | 2 | Two balls were removed from the basket. | correct | incorrect |

(c) Answer comparison by GPT4 and regex rule from OpenCompass for Asdiv-a dataset of Mistral-7B-Instruct Model

Figure 23: Case Study: Answer Comparison by GPT4 and Regex Rule from Various Datasets and Models.

