# OpenReview forum: "MathEval: A Comprehensive Benchmark for Evaluating Large Language Models on Mathematical Reasoning Capabilities"
_ICLR.cc/2025/Conference — ICLR 2025 Conference Withdrawn Submission_

### Official Review · Reviewer_1zuC · 2024-11-03

**Soundness:** 3
**Presentation:** 2
**Contribution:** 2
**Rating:** 3
**Confidence:** 3

**Summary:**

This paper presents a more comprehensive benchamrk for LLMs in the domain of maths, elaborates on how elicitations mechanisms (prompt engineering, chain of thought, et.c) should be applied to different families and then introduces automatic evaluation of the results via GPT-4 and a finetuned model. The experimental results are consistent with previous evidence. Nothing especially relevant is found from the experiments.

**Strengths:**

- Comprehensiveness, at least in languages and types of problems.
- Releasing a low-cost finetuned model (DeepSeek-7B) that can be used for automated evaluation can be very useful.
- Number of models and configurations being evaluated.

**Weaknesses:**

- The contribution is limited to the aggregation of benchmarks (some of them recently introduced with less contamination, but not evaluated separately) and automated grading.

- It's not clear whether the collection includes very difficult problems and may saturate soon. Current best results are around 70%, but not sure if there are some datasets or subdatasets where errors are quite high still. Aggregate results do not give a lot of insight.

- The prompt adaptation for models and datasets is not fully motivated. Is it the goal to evaluate each model and dataset in the condition that gets the best results? Is that fair and meaningful about an ecologically-valid use of these models in scenarios where maths knowledge is necessary? It's different to evaluate a specific LLMs for a competition and another is when these models are used in educational or engineering settings, to put two examples. In the end, why prompt adaptation instead of an off-the-shelf use? Language models should work in natural conditions, or all with similar china-of-thought prompts. In any case, how do we know that we use the optimal generic prompt for each LLMs?

Evaluation using LLMs is quite usual today. Section 2.3 tries to present this as new (but it's not even if multiple-choice problems are still very common), but there's significant work in this area:  https://arxiv.org/abs/2403.02839, https://arxiv.org/pdf/2406.18403, https://arxiv.org/abs/2408.02666 or https://eugeneyan.com/writing/llm-evaluators/.
Section 2.3 raises the qustion about how the evaluation from GPT-4 is validated. It seems this section is going to cover this, a comparison with a sample of answers evaluated by human experts? And it refers to appendix G.1 shows Table 7, but what's 0.6264 for instance? Inter-rater agreement? The caption says "overall average score". Are we choosing human annotations as ground truth? If this is agreement this is very low. What's the sample annotated by humans? The read may get confused because it seems this section is going to clarify what the quality of evaluation is by using GPT4. Then, this is expanded in section 3.2, and Figure 5 more specifically. But since the sample size is not mentioned, there's no sample and humans evaluate all the instances? What's the number of instances being labelled by each of the annotators? Using Kappa and having 0.8871 is good to use the human average (or mode?) as golden standard. Why for the 22 datasets or only 19? Why only 19? If this is done for 22, why figure 5 only with 19? Why not a sample of the 22? But then, in Figure 5, what does it mean to calculate an "absolute difference"? Are the answer other than correct and incorrect to calculate a "difference"? Why not Kappa as you did before? I cannot really determine whether the automated evaluations are good or bad as I cannot interpret the disagreement. And this is one of the key contributions of the paper.

Figure 3 and the related text is confusing. It's not clear whether all the things in blue and green are alwasy used, but not the one in black: "[COT prompt]". Is this optional? When is it introduced?

Is the configuration of prompts per model and dataset different? What if we need to explore a new model? Should we try to find the best prompts for each and every dataset in MATHBENCH?

The details about "Calculation Scheduling" and parallel processing are not part of the benchmarks, and definitely not part of the "prompts section". This is just experimental details or it could go to the appendix.

The evaluation results are based on an arithmetic mean of all datasets. This is a common practice but requires a justification, as the different datasets are incommensurate in difficulty. Why are easy datasets weighting the same as hard datasets? Do we have models failing at easy items but succeeding at difficult ones? Averages are not the best way of comparing systems. It is telling that the paper also reflects the results for GSM8K and MATH, and they see only minor discrepancies, so what's the point then about this comprehensive dataset if the same results could have been obtained with only GSM8K and MATH?

With this aggregate results, many of the observations in Fig. 6 are confirmatory, such as the best LLMs for maths (as we knew for some other dataset collections) are the best for this benchmark, and also the effect of finetuning, but specifically parameters (perhaps FLOPS would have been a better metric).

The separation between Math word problems and arithmetic in Fig. 6 (bottom) is more insightful, but the arithmetic variability is not explained (this is partly explained by "arithmetic plugins", but isolated benchmarks with basic operations using large numbers could have been conducted to know what models are using them or not).

The main contribution is an aggregation of datasets and some experimental results from them. The unification of prompting is questionable, especially in the way new models can be evaluated with this benchmark in an easy way, without the need for prompt adaptation.

Minor issues:
- Sometimes the authors use triple quotes, i.e., '''xxx''', which is non-standard.
- Conclusions, line 1: "In this papar"

**Questions:**

- Is prompting specialised for each pair of model and dataset?
- Is CoT used for some models but not others?
- How many instances did humans evaluate?
- Why 19 out of 22 for the comparison humans - automated scoring?
- What's the distribution of difficulty of the datasets and the evolution across that difficulty?
- Is there any unexpected finding in the experimental results?

---

> ### Author Response · Authors · 2024-11-28
>
> Firstly, we would like to express our sincere gratitude to the reviewers for their insightful feedback.
>
> ### Q1: The contribution is limited to the aggregation of benchmarks (some of them recently introduced with less contamination, but not evaluated separately) and automated grading.
> **A1**: MathEval offers a Comprehensive Evaluation Suite, which fills a crucial gap by providing a unified and thorough benchmark specifically designed to evaluate the mathematical reasoning capabilities of large language models (LLMs).
>
>  The significance of this contribution seems to have been overlooked, and we aim to clarify it. As shown in Figure 1, MathEval's Comprehensive Evaluation Suite consists of three parts: Math Scenarios, Prompt Adaptation, and LLM-based Evaluation. These correspond to the three issues identified in the Introduction: **incomprehensiveness, inadequate adaptation, and inconsistency**. Specifically: (1) Math Scenarios, which encompass languages (Chinese and English), problem types (arithmetic and math word problems), and educational levels (primary, middle, and high school), comprehensively address the challenge of **incomprehensiveness**; (2) Prompt Adaptation, which selects tailored datasets and model templates based on specific dataset characteristics and model information, effectively tackling the problem of **inadequate adaptation**; (3) LLM-based Evaluation, which utilizes GPT-4 for answer extraction and comparison to mitigate **inconsistency** issues, with an alternative distilled compare-answer model available for users without access to GPT-4.
>
> Thus, MathEval can leverage this Comprehensive Evaluation Suite to adapt to new datasets and models, providing a comprehensive and fair evaluation. This is the core of MathEval as a comprehensive benchmark for evaluation. The other contributions, such as expanding the dataset, performing multidimensional evaluation result analysis, and providing a Reliable Answer Extraction Model, are merely partial outcomes in the process of building MathEval.
>
> ### Q2: It's not clear whether the collection includes very difficult problems and may saturate soon. Current best results are around 70%, but not sure if there are some datasets or subdatasets where errors are quite high still. Aggregate results do not give a lot of insight.
> **A2:** Aggregate results indeed are indeed insufficient to give a lot of insight. Given the 52 models across 22 datasets, totaling 1,144 results, it is challenging to display all the data. Therefore, evaluation is conducted using high-level dimensions. In addition to the average score, Sections 3.4 and Appendix E present several analyses from different perspectives and uncover several intriguing insights. These include dimensions such as languages, problem types, educational levels, the differences between closed-source and open-source models, and the evaluation discrepancies arising from few-shot/zero-shot settings. Furthermore, the Gaokao-2023 results in Figure 13 on page 29 of the Appendix are used to discover potential data contamination.
>
> For further exploration of other perspectives or individual dataset-level analysis, a website will be provided to include all the specific evaluation results to support the discovery of more detailed findings. Specifically, all datasets currently covered are listed in Table 2. For particularly challenging datasets, this is an ongoing maintenance process, which includes the continuous expansion of the GAOKAO series evaluations and the newly added OlympiadBench evaluations. Current best results for these datasets are around 50% and 30%, respectively.

---

> > ### Author Response · Authors · 2024-11-28
> >
> > ### Q3: Regarding the issue of prompt adaptation
> > **A3**: Prompt adaptation is one of the main components of MathEval's Comprehensive Evaluation Suite, designed to ensure scalability in evaluating the 52 models across 22 datasets. When a new dataset is introduced, only a new dataset template needs to be set up, allowing it to be evaluated across the 52 models. Similarly, when a new model is introduced, only a new model template is required, enabling evaluation across the 22 datasets.
> >
> > **The design's primary goal** is to address the issues of "Inadequate adaptation" and "Inconsistency" mentioned in the Introduction. "Inadequate adaptation" refers to the need for a framework capable of handling the requirements of different types of models and datasets. For instance, chat models need to use special templates, while base models do not. This is highly sensitive and manually adjusting the evaluation code is impractical, necessitating an appropriate framework. "Inconsistency" addresses fairness concerns: for the same dataset, the prompt and cases used in few-shot evaluations should be fixed to avoid multiple leaderboard discrepancies. For example, OpenCompass, Llemma, and HELM use the same LLaMa2 model, but the accuracy on GSM8K and Math differs significantly: (16.7%, 3.3%), (11.8%, 3.2%), and (13.3%, 10.7%), respectively. For the same model, the prompt template should also be fixed to avoid results being affected by external factors.
> >
> > **Now, let’s respond to the specific questions:**
> >
> > ### Q3.1: The prompt adaptation for models and datasets is not fully motivated. Is it the goal to evaluate each model and dataset in the condition that gets the best results?
> > **A3.1**: As mentioned earlier, the goal is to ensure scalability while maintaining fairness. For datasets, this ensures uniform datasets configurations across different models, rather than primarily aiming to "get the best results." For evaluation models, fairness is ensured by using the specified template and system prompt. In general, when models are released, it is expected that the evaluation conditions are optimized for the best results, but this is not the task of the evaluation framework.
> >
> > ### Q3.2: Is that fair and meaningful regarding an ecologically-valid use of these models in scenarios where math knowledge is necessary? It's different to evaluate a specific LLM for a competition and another when these models are used in educational or engineering settings, to put two examples.
> > **A3.2**: This is why we categorize scenarios into three different dimensions. We want models to be chosen based on their suitability for different scenarios, such as elementary and high school situations, to ensure the results match the specific requirements of each setting.
> >
> > ### Q3.3: In the end, why prompt adaptation instead of an off-the-shelf use? Language models should work in natural conditions, or all with similar chain-of-thought prompts. In any case, how do we know that we use the optimal generic prompt for each LLM?
> > **A3.3**: This is directly related to the issues of "Inadequate adaptation" and "Inconsistency." Off-the-shelf use faces the problem of "Inadequate adaptation," which makes the evaluation framework non-scalable. "Inconsistency" leads to unfairness. Only model-specific prompts aim for the optimal results, but this is akin to model parameters, which are finalized when the model is released, rather than discovered during evaluation. On the other hand, the shots used for datasets should be standardized.

---

> > > ### Author Response · Authors · 2024-11-28
> > >
> > > ## Q4: Evaluation methods details
> > > ### Q4.1: Section 2.3 raises the question about how the evaluation from GPT-4 is validated. It seems this section is going to cover this, a comparison with a sample of answers evaluated by human experts?
> > > **A4.1**: Section 2 introduces MathEval's Comprehensive Evaluation Suite, which includes three components: Math Scenarios, Prompt Adaptation, and LLM-based Evaluation (see Figure 1). Section 2.3 specifically discusses the LLM-based Evaluation method. It covers two interchangeable methods: 1) GPT-4 for answer extraction and comparison, and 2) an alternative distilled compare-answer model. These methods serve as alternatives to ensure a robust and fair evaluation. Human experts are not involved in this section; they are only engaged in evaluating the effectiveness of these two evaluation methods in Section 3.2.
> > >
> > > ### Q4.2: And it refers to appendix G.1 shows Table 7, but what's 0.6264 for instance? Inter-rater agreement? The caption says "overall average score." Are we choosing human annotations as ground truth? If this is agreement, this is very low. What's the sample annotated by humans? The reader may get confused because it seems this section is going to clarify what the quality of evaluation is by using GPT-4.
> > > **A4.2**: There seems to be a misunderstanding. Table 7 is found in Section 3.2 (G.3), not in Section 2.3 (G.1).
> > >
> > > Table 7 compares the overall average score across four models using three Compare Answer Methods on all datasets. The three methods listed in the first column of the table are: Human Annotated, Two-stage with GPT-4, and Fine-tuned-DeepSeek-7B. The four models include GPT-4, DeepSeek-math-7B-Base, DeepSeek-math-7B-Instruct, and DeepSeek-math-7B-RL, representing closed-source and various open-source models (Base, Instruct, and RL).
> > >
> > > The score of 0.6264 represents the overall average score across all datasets for Human Annotated GPT-4 evaluations.
> > >
> > > **The primary purpose of Table 7** is to validate the reliability of the Two-stage with GPT-4 and Fine-tuned-DeepSeek-7B Compare Answer Methods based on the assumption that human annotations are accurate and can be used as ground truth.
> > >
> > > ### Q4.3: Then, this is expanded in Section 3.2, and Figure 5 more specifically. But since the sample size is not mentioned, there's no sample and humans evaluate all the instances? What's the number of instances being labeled by each of the annotators? Using Kappa and having 0.8871 is good to use the human average (or mode?) as a golden standard.
> > > **A4.3**: You are correct in your understanding; this part verifies the accuracy of human annotations. Using Kappa and achieving 0.8871 indicates the consistency of human annotations in determining correctness, which reflects inter-annotator agreement.
> > > Specifically, we conducted human annotations on the outputs of the four models listed in Table 7. Each model's outputs across 19 datasets, evaluated under both zero-shot and few-shot settings, resulted in a total of 53,400 outputs. Each output was labeled by 5 annotators. Therefore, the total number of annotations was: 4 models * 53,400 outputs * 5 annotations, resulting in 1,068,000 annotations.
> > >
> > > ### Q4.4: Why for the 22 datasets or only 19? Why only 19? If this is done for 22, why is Figure 5 based only on 19? Why not a sample of the 22?
> > > **A4.4**: This is a detail related to dataset maintenance. MathEval is a continually evolving suite, and at the time of planning the human annotations, MathEval included 19 datasets. Later, OlympiadBench-CN, OlympiadBench-EN, and GAOKAO-2024 were added, expanding the collection to 22 datasets. Since the purpose of the human annotations was primarily to validate the effectiveness of the Compare Answer Methods, and not for the model performance evaluations or leaderboard rankings, we did not extend the annotations to the newly added datasets. Thus, the human annotation process was based on the 19 datasets originally included.

---

> ### Author Response · Authors · 2024-11-28
>
> ### Q5: Figure 3 and the related text is confusing. It's not clear whether all the things in blue and green are alwasy used, but not the one in black: "[COT prompt]". Is this optional? When is it introduced?
>
> **A5**: Referring to Figure 3, Figure 3(a) represents two independent processes: Model and Dataset Configuration. An example of this process is provided in Figure 19. In Figure 3(b), the prompts from Model and Dataset Configuration are wrapped for use in zero-shot and few-shot settings, with examples shown in Figures 20 and 21.
>
>  Specifically, the Dataset Configuration prompt templates include DQP, DAP, DOP, and COT prompts. The first three are clearly indicated in Figure 3(b) with their positions for concatenation. The COT prompt is more specific, as its placement depends on the dataset. Therefore, it is difficult to indicate this in the figure.
>
>  The use of the COT prompt is another issue. In a zero-shot setting, the COT prompt is always used, but in a few-shot setting, some datasets do not have a COT process, such as arithmetic problems. In those cases, the shots provided do not involve a COT process, meaning the COT prompt may not be applied. The use of [COT prompt] is therefore optional and depends on the dataset. The [] signify that this is conditional based on the dataset.
>
> ### Q6: Is the configuration of prompts per model and dataset different? What if we need to explore a new model? Should we try to find the best prompts for each and every dataset in MATHBENCH?
>
> **A6**: As mentioned in A3, when introducing a new model, the only thing that needs to be adjusted is the new model template, which is generally based on the prompt provided upon the model’s release.
>
> ### Q7: The details about "Calculation Scheduling" and parallel processing are not part of the benchmarks, and definitely not part of the "prompts section". This is just experimental details or it could go to the appendix.
>
> **A7**: Calculation Scheduling is indeed not directly related to prompt adaptation, but it is part of the answer generation process, as illustrated in Figure 3. It is an important aspect of the practical implementation of a benchmark. Reviewers, such as Reviewer Nkro, may focus on its specific implementation details.
>
> ### Q8: The evaluation results are based on an arithmetic mean of all datasets. This is a common practice but requires a justification, as the different datasets are incommensurate in difficulty. Why are easy datasets weighting the same as hard datasets? Do we have models failing at easy items but succeeding at difficult ones? Averages are not the best way of comparing systems. It is telling that the paper also reflects the results for GSM8K and MATH, and they see only minor discrepancies, so what's the point then about this comprehensive dataset if the same results could have been obtained with only GSM8K and MATH?
>
> **A8**: This addresses two issues. First, the arithmetic mean is indeed a single metric. As explained in A2, we have also performed analysis across three dimensions, including Grade, and provided the complete evaluation results for further analysis. Secondly, the results for GSM8K and MATH, referenced in Appendix F.3 (Table 6), are meant to help with the development of the Evaluation Suite and validate the accuracy of the evaluation, not to suggest that these benchmarks alone are sufficient.
>
> ### Q9: With this aggregate results, many of the observations in Fig. 6 are confirmatory, such as the best LLMs for maths (as we knew for some other dataset collections) are the best for this benchmark, and also the effect of finetuning, but specifically parameters (perhaps FLOPS would have been a better metric).
>
> **A9**: Thank you for the suggestion. In addition to the confirmatory observations, we also identified some additional findings, which are discussed in Section 3.4 and Appendix E. For example, newer model series exhibit steeper slopes, indicating that their mathematical abilities improve more effectively with an increase in parameter size.
>
> ### Q10: The separation between Math word problems and arithmetic in Fig. 6 (bottom) is more insightful, but the arithmetic variability is not explained (this is partly explained by "arithmetic plugins", but isolated benchmarks with basic operations using large numbers could have been conducted to know what models are using them or not).
>
> **A10:** Thank you for the suggestion. This could potentially lead to a new dataset design, and we are considering adding this to MathEval in the future.

---

> > ### Author Response · Authors · 2024-11-28
> >
> > ### Q11: The main contribution is an aggregation of datasets and some experimental results from them. The unification of prompting is questionable, especially in the way new models can be evaluated with this benchmark in an easy way, without the need for prompt adaptation.
> > **A11**: This is similar to the issue raised in Q3. We still want to emphasize that the evaluation of 52 models across 22 datasets is scalable and fair, and this scalability and fairness are enabled by prompt adaptation.
> >
> > ### Q12: Minor issues
> > **A12**: Thank you for pointing these out. We have corrected them in the new PDF version.
> >
> > ---
> >
> > ### Q13: Is prompting specialized for each pair of model and dataset?
> >
> > **A13**: Yes, prompting is specialized for each pair of model and dataset. There are 22 dataset configurations and 52 model configurations, resulting in 22*52 specialized prompts. Each prompt is evaluated in both zero-shot and few-shot settings. Specifically, Figure 3 illustrates the prompt adaptation process. Figure 3(a) shows two independent processes for Model and Dataset Configuration, and an example is provided in Figure 19. In Figure 3(b), the prompts from (a) are wrapped based on zero-shot and few-shot settings, with examples shown in Figures 20 and 21.
> >
> > ### Q14: Is CoT used for some models but not others?
> > **A14:** CoT is part of the dataset configuration. For some datasets, the answers do not involve a CoT process, so CoT is not used in few-shot settings for those datasets. However, in zero-shot settings and for other datasets, CoT is used.
> >
> > ### Q15: How many instances did humans evaluate?
> > **A15**: Humans evaluated approximately 1,068,000 instances. More details can be found in the previous response A4.3.
> >
> > ### Q16: Why 19 out of 22 for the comparison of humans vs. automated scoring?
> > **A16**: MathEval is a continually evolving suite, and at the time of planning the human annotations, MathEval included 19 datasets. Later, OlympiadBench-CN, OlympiadBench-EN, and GAOKAO-2024 were added, expanding the collection to 22 datasets. Since the purpose of the human annotations was primarily to validate the effectiveness of the Compare Answer Methods, and not for the model performance evaluations or leaderboard rankings, we did not extend the annotations to the newly added datasets. Thus, the human annotation process was based on the 19 datasets originally included.
> >
> > ### Q17: What's the distribution of difficulty of the datasets and the evolution across that difficulty?
> > **A17**: We have added an analysis of the dataset distribution in Appendix B.2. As shown in Figure 9, by applying t-SNE to query embeddings from the 22 datasets and visualizing the results, we observe that the datasets naturally form three clusters: English datasets, Chinese datasets, and arithmetic datasets. Further examination of the Chinese and English clusters shows that as the t-SNE component 2 value decreases (from top to bottom in the figure), the problems become progressively more difficult, and the corresponding grade levels also rise. This naturally reflects that our difficulty levels are distributed across various grade levels, with datasets corresponding to each level.
> >
> > ### Q18: Is there any unexpected finding in the experimental results?
> > **A18**: In Section 3.4 and Appendix E, along with some high-level analyses, we also included a discussion on potential data contamination. We compared the model rankings between Gaokao-2023 and the overall average score. Since Gaokao-2023 is a brand-new set of questions, we wouldn't expect significant variations in rankings if there were no contamination. Therefore, substantial differences in ranking suggest potential data contamination.
> >
> > We found that the Qwen-series models might have encountered such contamination, and chat models are more likely to have been exposed to similar math word problems during the instruction fine-tuning stage, increasing the probability of data contamination. Further discussion on this can be found in our response to Reviewer kkq7's fourth part.
> >
> > ---
> >
> > **Finally**, we thank the reviewers for their detailed feedback. We would like to reiterate the importance of prompt adaptation for MathEval as a benchmark. In addition to the discussions above, we provide an  [anonymous github](https://anonymous.4open.science/r/MathEval-505B/README.md) for a better understanding of the implementation process and the significant role of prompt adaptation.

---

> ### Author Response · Authors · 2024-12-02
> **Have we addressed your concern**
>
> Thank you very much for your thorough review. As the discussion period is nearing its conclusion, I wanted to follow up to ensure you’ve had the opportunity to review our detailed rebuttal. Given the additional explanations and adjustments we've incorporated, we would greatly appreciate your feedback on whether our responses have satisfactorily addressed your concerns.
>
> Thank you once again for your time and thoughtful review. We look forward to your response.

---

### Official Review · Reviewer_kkq7 · 2024-11-04

**Soundness:** 3
**Presentation:** 3
**Contribution:** 1
**Rating:** 3
**Confidence:** 3

**Summary:**

This paper introduces MathEval, a unified evaluation suite for mathematical reasoning of LLMs. It puts together 22 datasets, and provides a unified method to extract and score answers given model responses. The base answer extraction method in MathEval uses GPT-4, but the authors provide a fine-tuned DeepSeek-Math 7B model just for the task of answer extraction. Authors provide experiments on 52 closed and open models, with Claude 3.5 Sonnet performing best overall in both English and Chinese.

**Strengths:**

* The paper is easy to read
* The unified dataset is likely to be useful for the community. Answer extraction is indeed a pain for math evaluations, and it's useful to have a standardized method / model for this. I believe people would use this.

**Weaknesses:**

* While I do think the effort behind MathEval is useful, I'm not sure if it provides sufficient insight or novelty for an ICLR research paper. I did not gain any insights from the paper, and by construction it is composed of existing evaluations of reasoning capabilities.
* It's unclear to me how much overlap there is in the datasets in terms of distribution of problems. Besides the multilinguality, I'm not sure the datasets are measuring fundamentally different things.
*

**Questions:**

* Have the authors looked at correlations between performance on all the 22 datasets?
** If this correlation is high (which I'd suppose it is), what is the gain in using so many datasets, since their performance can be predicted from one another?
** If this correlation is low between some pairs, what orthogonal abilities might they be measuring?
* What insights did the authors derive from running all the evaluations that would not be obvious from just a small subset of them (e.g. one from each category in Fig 2)?

**Details Of Ethics Concerns:**

* Missing details on how the human annotators were recruited / compensated. It's unclear if they're authors of the paper, or were hired.
* If the data of all the datasets is to be released in MathEval, it should be checked that their license allows this.

---

> ### Author Response · Authors · 2024-11-26
>
> We sincerely thank the reviewer for the thoughtful feedback and insightful questions. We are pleased to hear that you find our paper easy to read and acknowledge the utility of MathEval for the community
>
> **1, Regarding the novelty and contribution of MathEval:**
>
> We appreciate reviewer's concern about the novelty aspect of our work for an ICLR research paper. While we acknowledge that benchmarks may not always introduce novel methodologies, we believe that the contributions of MathEval are significant for the following reasons:
>
> - Comprehensive Evaluation Suite: MathEval fills a crucial gap by providing a unified and comprehensive benchmark specifically designed for evaluating the mathematical reasoning capabilities of large language models (LLMs). It encompasses 22 diverse datasets that cover various problem types (e.g., computation, application, multiple-choice), difficulty levels (elementary to high school), and languages (English and Chinese).
> - Reliable Answer Extraction Model: We address the substantial challenge of answer extraction in mathematical problem-solving by introducing a fine-tuned DeepSeek-Math 7B model. This model enhances the reliability and consistency of evaluations, which is essential for fair model comparisons.
> - Insights into LLM Performance: By conducting extensive experiments on 52 models, we provide valuable insights into the strengths and weaknesses of current LLMs in mathematical reasoning across different dimensions, including language proficiency, grade levels, and potential data contamination.
>
>
> Furthermore, as benchmarks like [MathVista](https://openreview.net/pdf?id=KUNzEQMWU7) and [BooookScore](https://openreview.net/pdf?id=7Ttk3RzDeu) have demonstrated in prior ICLR publications, the contribution of a well-designed benchmark lies in its ability to propel the community forward by providing a reliable and robust platform for evaluation. We believe MathEval will be instrumental for future research in this domain.
>
>
> **2. Regarding the potential overlap among datasets and the aspects they measure:**
>
> We appreciate reviewer's insightful concern about the potential overlap in our datasets and whether they are truly measuring different aspects of mathematical reasoning beyond multilinguality.
>
> To address this, we have taken several steps in our revised submission:
>
> - Detailed Dataset Classification (Appendix B.2, Table 2): We have included a comprehensive table in Appendix B.2 (pages 16-17) that lists each dataset along with its specific classification.
> - Analysis of Potential Overlap Between Datasets- (Page 16, 17, Line 711-731, Appendix B.2):
>     - Query Similarity Heat Map (Figure 8): To examine the overlap between datasets, we conducted a query similarity analysis using embedding techniques to represent each problem query. We computed pairwise cosine similarities between queries from different datasets and visualized the results in a heat map (Figure 8). The heat map illustrates that the average similarities between queries from different datasets are low, indicating minimal overlap in problem content.
>     - Dimensionality Reduction and Clustering (Figure 9): Furthermore, we applied dimensionality reduction (e.g., t-SNE or UMAP) to the query embeddings and performed clustering to visualize the distribution of queries across datasets (Figure 9). The resulting plot shows that queries from different datasets form distinct clusters, suggesting that each dataset covers unique topics and problem types.
> - Findings from the Analysis - (Page 16, 17, Line 711-731, Appendix B.2):
>     - Coverage of Diverse Difficulty Levels: Our datasets collectively cover a wide range of difficulty levels, from elementary school mathematics to high school competition-level problems. This ensures a comprehensive assessment of models across varying complexities.
>     - Low Inter-Dataset Similarity: The low similarity scores between queries from different datasets confirm that each dataset presents unique challenges. This indicates that the datasets are measuring fundamentally different aspects of mathematical reasoning, such as basic computation, complex problem-solving, logical reasoning, and application of mathematical concepts in various contexts.
>
>
> By conducting this detailed analysis and including these findings in our revised paper, we aim to demonstrate that the datasets within MathEval are not overlapping but are instead complementary, each contributing to a holistic evaluation of mathematical reasoning in LLMs.

---

> ### Author Response · Authors · 2024-11-26
>
> **3. Addressing the correlations between performances on all 22 datasets:**
>
> We appreciate your insightful suggestion. In response, we have conducted a comprehensive correlation analysis, which is presented in Figure 10 on page 19 (lines 850-861) of our revised submission.
>
> - Findings from the Correlation Analysis (Figure 10): Our analysis shows that there are indeed high correlations between model performances on many of the datasets. We consider this to be a reasonable and expected outcome because mathematical abilities are interconnected. Improvements in computational skills often enhance problem-solving capabilities, and advancements in reasoning skills typically benefit performance across various mathematical tasks. This interrelated nature of mathematical competencies suggests that as models improve in one area, they tend to improve in others as well. Which proves our benchmark is reliable.
>
> - Justification for Including Multiple Datasets:
>     - Enhancing Benchmark Robustness: Including all 22 datasets increases the robustness and reliability of our benchmark. A comprehensive evaluation across diverse datasets ensures that models are not just performing well on a narrow set of problems but are demonstrating consistent mathematical reasoning abilities across a wide spectrum of topics and difficulty levels.
>     - Identifying Specific Strengths and Weaknesses: Even with high overall correlations, the detailed performance on individual datasets can reveal specific strengths or weaknesses of a model.
>     - Detecting Potential Data Contamination: A significant benefit of using multiple datasets is the ability to detect potential data contamination. For example, suppose Model M has high performance on Dataset A but does not show similar performance gains on other highly correlated datasets. This discrepancy might suggest that Model M has potentially memorized Dataset A due to data leakage, rather than genuinely understanding the underlying mathematical concepts.
> - Measuring Orthogonal Abilities: While correlations are generally high, our analysis also reveals that certain datasets measure more specialized or orthogonal abilities. For instance, some datasets focus on advanced reasoning or problem-solving skills that may not correlate as strongly with basic computational abilities. Including these datasets allows us to assess a wider range of mathematical proficiencies.
>
> **4. Insights derived from the comprehensive evaluations:**
>
> By conducting comprehensive evaluations across all datasets, we uncovered insights that wouldn't have been apparent from analyzing just a small subset. One significant discovery was the potential data contamination identified in Figure 13 on page 29 of the Appendix.
>
> This figure was instrumental because it highlighted discrepancies in model performance on the Gaokao-2023 dataset—a brand-new set of questions that none of the models had encountered during training. Given the high correlation in performance across datasets (as discussed in our previous answer), we wouldn't expect significant variations on Gaokao-2023 if there were no contamination. Therefore, substantial differences in rank suggest potential data contamination.
>
> In the Upper Chart of Figure 13:
> - Chinese Subsets Rank (Blue Bars): Indicates each model's ranking within Chinese mathematical datasets. A smaller rank signifies better performance.
> - Gaokao-2023 Rank Increase (Orange Bars): Represents models whose rank increased (i.e., performed worse) on Gaokao-2023 compared to other datasets. A larger increase indicates poorer performance on Gaokao-2023.
> - Gaokao-2023 Rank Decrease (Green Bars): Represents models whose rank decreased (i.e., performed better) on Gaokao-2023. A larger decrease signifies better performance on the new dataset.
>
> In the Lower Chart:
>
> Similar to the upper chart, but the blue bars represent the overall average score across all 22 datasets.
>
> From the figure, we detected potential data contamination:
>
> - The top two models showing a significant increase in rank (poorer performance) on Gaokao-2023 are ChatGLM3-6B and Baichuan2-13B.
> - Many of the Qwen-series models display orange bars, suggesting they may have been trained on data overlapping with our evaluation sets, leading to inflated performance on those but not on Gaokao-2023.
>
> These observations are further supported by findings in the paper "Compression Represents Intelligence Linearly", which discusses similar issues for Qwen-series model. Furthermore, most base models exhibit green bars, which suggests that chat models are more likely to have encountered similar math word problems during the instruction fine-tuning stage, increasing the probability of data contamination.

---

> ### Author Response · Authors · 2024-12-02
> **Have we addressed your concern**
>
> Thank you very much for your thorough review. As the discussion period is nearing its conclusion, I wanted to follow up to ensure you’ve had the opportunity to review our detailed rebuttal. Given the additional explanations and adjustments we've incorporated, we would greatly appreciate your feedback on whether our responses have satisfactorily addressed your concerns.
>
> Thank you once again for your time and thoughtful review. We look forward to your response.

---

> > ### Comment · Reviewer_kkq7 · 2024-12-02
> >
> > I thank the authors for the very through response. I looked at all the updates to the manuscript, including in the appendix, which I think are very informative.
> >
> > > 1, Regarding the novelty and contribution of MathEval:
> >
> > Methodologically, I completely see the value of the answer extraction model that is introduced here: I believe it has potential to be used in future work.
> >
> > However, I'm still unconvinced by what exactly MathEval as a benchmark suite enables. To quote from the authors response:
> >
> > >> MathEval fills a crucial gap by providing a unified and comprehensive benchmark specifically designed for evaluating the mathematical reasoning capabilities of large language models (LLMs)
> >
> > >> By conducting extensive experiments on 52 models, we provide valuable insights into the strengths and weaknesses of current LLMs in mathematical reasoning across different dimensions
> >
> > What I still fail to see is what is this crucial gap, or what are the valuable insights, concretely. I'm obviously not opposed to benchmarking papers in ICLR -- many of these papers can indeed help guide the community towards new interesting directions. I think the two examples that the authors cite -- MathVista and BooookScore -- were examples of this. MathVista became a standard evaluation of visual reasoning, which was (in a general setting) a new task, in response to the very recent release to GPT-4V. The datasets that comprised MathVista were extremely specialized, so that putting them together indeed produced a new whole. They also measured the human evaluation gap, which was useful to set new goals for the community. For BooookScore, that was one of the first evaluation of very long context language tasks, only recently supported by LLMs and expected to grow. For MathEval, I'm not seeing what is the equivalent novel task or evaluation angle that is gained by putting all the existing benchmarks together.
> >
> > > 2. Regarding the potential overlap among datasets and the aspects they measure:
> >
> > Thanks, this is helpful (more the textual descriptions than the cosine-based analyses - those are hard to interpret because it's hard to know what the embedding model is most sensitive to -- syntax? wording? formatting? mathematical semantics? these all could cause clusters besides the mathematical content itself).
> >
> > > 3. Addressing the correlations between performances on all 22 datasets:
> >
> > Thank you, I think this is extremely helpful to understand the dataset.
> >
> > The correlations indeed generally look quite high. It seems like there are basically two clusters here: asdiv-a has the weakest correlations with the rest, while most of the other correlations seem to be generally >= .7, with few exceptions. This would indicate that, if one were to just pick a few of the datasets, they would already be very good proxies for the rest (which is the current practice).
> >
> > > 4. Insights derived from the comprehensive evaluations:
> >
> > This is an interesting analysis. It might be a lead into something interesting, but it's still more on the speculative side rather than a clear finding (since you can't really test this by looking into the data for most models, unfortunately).
> >
> > Looking at the ranks is a bit opaque, because if many models are performing very close to each other, then a small difference in performance might cause a large difference in rank. Besides, even if rank stays the same but the top-1 model drops in accuracy by a lot, this would already be interesting (and conversely, if all ranks change a lot but overall performances are within a few % of their originals, this wouldn't be so surprising).
> >
> > Overall, I think the potential data contamination analysis is enabled by the new GAOKAO datasets that are introduced (which are a nice contribution), rather than necessarily putting all of the other datasets together as the MathEval suite proposes to do. Perhaps there could be more focused and clear insights that can be derived from the new datasets, instead of focusing on the overall evaluation from which I did not see a concrete takeaway.

---

### Official Review · Reviewer_BH88 · 2024-11-04

**Soundness:** 4
**Presentation:** 3
**Contribution:** 3
**Rating:** 6
**Confidence:** 5

**Summary:**

The authors present a benchmark that aims to address three dimensions of current math benchmarks: comprehensiveness, adaptation, and consistency. MathEval is a benchmark specifically designed to evaluate the mathematical reasoning abilities of LLMs across problem types, languages, and difficulty levels, encompassing primary through high school math problems in English and Chinese. The benchmark includes 22 datasets and integrates a dynamic update feature, adding new problems annually to reduce test data contamination. MathEval uses a tailored prompting approach to adapt to the unique characteristics of different models and problem types, ensuring fairer and more accurate comparisons. To maintain consistency and overcome the limitations of traditional rule-based evaluation methods, the benchmark uses GPT-4 for answer extraction and comparison, with a publicly available deepseek-7B model as an accessible alternative.

**Strengths:**

* The authors create a comprehensive benchmark that includes 22 math reasoning benchmarks, some of which they create. The benchmark is annotated with the following dimensions: language of the problem, educational level (primary to high school), and problem type (arithmetic vs. word problem).
* Because different models and different datasets require unique prompting techniques, the paper also includes adaptable prompt templates which make it easier to evaluate a model under zero/few shot conditions depending on what is more suited for the model.
* The paper also presents an open model that can be used to compare mathematical answers for researchers who might not have access to GPT-4 to use for grading.

**Weaknesses:**

* Some previous work (https://arxiv.org/pdf/2407.00900) has been done around using open source LLMs as part of the grading framework, although fine tuning a specific model for the answer comparison task is still a notable contribution.
* While the paper presents a significant effort in benchmarking (and the tools presented to the broader research community via this paper will be useful to researchers), the discussion from the numerical results support a lot of things that are already well known (the supremacy of closed-source over open-source models, the performance of math domain models generally being better, few-shot prompting generally resulting in better performance when compared to zero-shot, etc.)

**Questions:**

* To minimize compute costs, wouldn’t it make more sense to first try regex-based answer extraction on an answer, and in the case that the regex-extracted answer is incorrect (which could be caused by either a genuinely incorrect answer, or a mis-extracted answer), run GPT-4/the custom model on the answer? Because in Figure 14, we see that precision for answer verification with regex-only is high enough that GPT-4 doesn’t need to be run on every single model output, and rather only on ones that are originally marked as incorrect.
* In lines 415-418, the authors note that the dataset-level higher setting consistency outperforms using either few- or zero-shot prompting. However, isn’t dataset-level higher defined as the higher accuracy between few- and zero-shot accuracy at the dataset level? So wouldn’t this behavior be expected by definition? Sorry if I am misunderstanding the definition.
* Just a quick note about the conclusion section (line 464): the paragraph mentions 2 datasets when MathEval actually includes 22.

---

> ### Author Response · Authors · 2024-11-20
>
> Firstly, we would like to express our sincere gratitude to the reviewers for their insightful feedback.
>
> ### Previous Work on Using Open Source LLMs in Grading Frameworks:
>
> Thanks reviewer for highlighting the relevant work MathCAMPS (https://arxiv.org/pdf/2407.00900), which uses large language models for problem rephrasing and symbolic solvers for answer verification. While MathCAMPS focuses on generating diverse datasets, our approach stands out by addressing more natural and realistic problems across a range of difficulty levels from elementary to high school. We appreciate the reviewer's recognition of our efforts in fine-tuning a specific model for the answer comparison task.
>
> ### Well-Known Findings Discussion:
>
> We recognize that some of our discussions cover well-known findings within the domain. Due to space constraints, a more extensive discussion is included in Appendix E (Lines 824-952), where we conduct a detailed analysis that may offer more valuable insights for researchers than the content in the main text. We summarize the content as follows:
>
> - **Language Dimension:** Models generally exhibit stronger mathematical performance in English than in Chinese, especially at the primary school level. This disparity is attributed to primary school math problems requiring more language comprehension, and the models trained predominantly on English datasets lack sufficient exposure to Chinese mathematical problems. Models developed by Chinese companies (e.g., WenXin 4.0, Spark-3.5) perform better in Chinese due to their training data, while those from English-speaking countries excel in English.
>
> - **Impact of Specialized Fine-Tuning:** Fine-tuning models with specialized mathematical data (e.g., MAmmoTH-70B, MetaMath-70B) significantly enhances their problem-solving abilities, highlighting the importance of domain-specific fine-tuning in boosting performance beyond specific datasets.
>
> - **Grade Level Dimension:** Models generally perform better on primary school math problems than on high school ones due to difficulty differences. Models like Claude-3.5-Sonnet and Gemini-1.5-Pro excel at the primary level, suggesting strong language comprehension that aids in solving word problems. In contrast, models like Llemma-7B and Llemma-34B show less pronounced advantages, possibly because their training focuses on complex concepts relevant to higher grades.
>
> - **Consistency Across Dimensions:** Models often exhibit consistent performance within the same dimension; strong performance in one language or grade level typically correlates with similar performance in related areas. Evaluating models across different dimensions is crucial for identifying specific strengths, weaknesses, and potential data contamination. Significant discrepancies—such as exceptional performance at one grade level but poor performance in other tasks—may indicate contamination with data from that particular grade level.
>
> We believe these detailed analyses provide a deeper understanding and help the community make informed decisions on model selection based on specific tasks.
>
> Moreover, on page 29 of the appendix, in Figure 13, we present potential test set contamination for the evaluated model.
>
> In the **Upper Chart**, we have three types of bars:
>
> - Chinese Subsets Rank (Blue Bars): This indicates how each model ranks specifically within Chinese mathematical datasets. A smaller rank indicates better performance.
>
> - Gaokao-2023 Rank Increase (Orange Bars): Represents increases in the rank of models when evaluated using Gaokao-2023 tests. A larger increase in rank signifies poorer performance on Gaokao-2023.
>
> - Gaokao-2023 Rank Decrease (Green Bars): Represents decreases in the rank of models with the Gaokao-2023 tests. A larger decrease in rank signifies better performance on Gaokao-2023.
>
> In the Lower Chart, we have the same three types of bars as in the upper chart; however, the blue bars represent the overall average score across 22 datasets.
>
> We believe that Gaokao-2023 is not contained in the training dataset for all models. Thus, if a model performs very well on the blue bars but poorly on Gaokao-2023, this may indicate potential test data contamination.
>
> From this figure, we can detect potential data contamination in the results. In the upper chart, the top two models showing an increase in rank are chatglm3-6b and Baichuan2-13b. Most of the Qwen-series models also have orange bars, indicating potential data contamination for these models. This conclusion is supported by findings in the paper "Compression Represents Intelligence Linearly". Furthermore, most base models exhibit green bars, indicating improved rankings on the \textit{Gaokao-2023} dataset. This suggests that chat models are more likely to have encountered similar math word problems during the instruction fine-tuning stage, increasing the probability of data contamination.

---

> > ### Comment · Reviewer_BH88 · 2024-12-03
> >
> > Thank you for updating your paper with the additional details, as they help support the work. However, I believe my current score reflects my confidence in the work. Thank you again for addressing the other issues.

---

> ### Author Response · Authors · 2024-11-20
>
> ## About Question1:
>
> We appreciate the suggestion to use a regex-based extraction as a first step to minimize compute costs. While this approach could potentially reduce the number of instances where a more complex model like GPT-4 is needed, our experiments indicate that relying solely on regex extraction can lead to potential issues, especially for certain types of tasks.
>
> As shown in Figure 14 of our paper, the regex-based method performs poorly on datasets like MathQA and MATH401, which involve mathematical word problems and arithmetic problems, respectively. In these cases, the complexity of the language and the variety of valid answer formats often lead to incorrect extractions or false positives—where an incorrect answer might be mistakenly marked as correct due to misextraction. This undermines the fairness and accuracy of the benchmark evaluation.
>
> To address both the computational cost and the need for reliable verification across diverse tasks, we have developed a compare-answer model with only 7 billion parameters. This model strikes a balance by significantly reducing the computational overhead compared to GPT-4 while providing more consistent and accurate answer verification than regex-based methods.
>
> ## About Question2:
>
> Your understanding is correct. The dataset-level higher setting is defined as selecting the higher accuracy between few-shot and zero-shot prompting. Our intention in highlighting this result was to emphasize the fairness and robustness of our evaluation. Evaluating a model using only few-shot or only zero-shot prompting can lead to an underestimation of the model's true capabilities on certain tasks. By considering the higher accuracy from either prompting strategy on a per-dataset basis, we provide a more balanced and fair assessment of the model's performance.
>
>
>
> ## About Question3:
>
> Great thanks to the reviewer for pointing out this typo, we will fix this in the revised version of the paper.

---

> ### Author Response · Authors · 2024-11-26
>
> Regarding Weakness 2:
>
> We have expanded our discussion on Pages 22-23 to address this concern more comprehensively. Specifically, we have included an analysis of potential data contamination and its implications for our study. Additionally, we have refined some of our previous conclusions to provide deeper insights and to strengthen the validity of our findings.
>
> Regarding Question 2:
>
> We have made minor revisions on Page 9, Lines 416-419, to clarify this point.
>
> Regarding the typos in our conclusion:
>
> Thank you for pointing out these errors. We have corrected them accordingly to enhance the readability and professionalism of our manuscript.

---

### Official Review · Reviewer_7NzK · 2024-11-04

**Soundness:** 2
**Presentation:** 3
**Contribution:** 3
**Rating:** 6
**Confidence:** 2

**Summary:**

MathEval represents a comprehensive benchmark designed to rigorously evaluate the mathematical problem-solving skills of large language models (LLMs).The benchmark is stratified by language, problem category, and difficulty, ensuring a comprehensive evaluation.To standardize responses and ensure consistent evaluation criteria, an automated LLM-driven pipeline for answer extraction and comparison has been developed.MathEval provides an extensive benchmark that includes a diverse array of mathematical problems across different types and difficulty levels. MathEval have developed a standardized method for comparing answers that effectively addresses the complexities associated with outputs from mathematical word problems (MWPs).MathEval implements a strategy of using a dynamically updated dataset.

**Strengths:**

This proposed framework is really useful and provides a more general evaluation standard for the evaluation of mathematical reasoning ability of subsequent large language models.And the problems it can solve are also diversified.

**Weaknesses:**

1. For a more mathematical process, it seems that a better evaluation standard cannot be given;
2. For the multi-step mathematical reasoning process, it seems impossible to evaluate whether this process is moving towards the expected solution process.
3. The classification of mathematical reasoning evaluated in the experiment is still a little rough.

**Questions:**

1. Can this framework evaluate the reasoning process? Because sometimes the results are the same, the complexity of the reasoning process is also a point that needs to be considered in the reasoning evaluation of large language models.

---

> ### Author Response · Authors · 2024-11-20
>
> Thanks reviewer for the thoughtful review, we would like to address the concrens and questions you raised:
>
> ## For Weakness 1:
>
> We understand that the reviewer is pointing out that our evaluation standard may not be sufficient for assessing the mathematical process itself. In our current framework, we choose to directly evaluate whether the final output of the model is correct. This decision was made because evaluating the solution process is a complex issue that we have considered extensively.
>
>
> Recent methodologies, such as Program Reward Models (PRM) or those based on attention score distribution for hallucination detection, have not yet demonstrated reliable accuracy in evaluating reasoning steps. Incorporating such methods into our benchmark could introduce unnecessary variables and potentially undermine the credibility of our results due to potential misjudgments.
>
>
> Our current approach focuses on the final answer verification, which we believe offers a more straightforward and reliable measure of performance. However, we recognize the importance of evaluating the reasoning process and are actively exploring this as a research topic. It remains one of our objectives to develop and integrate robust methods for reasoning process verification in the future.
>
>
> ## For weakness 2 and Question 1:
>
> Reviewer bring up an important point regarding the evaluation of multi-step reasoning processes. Currently, our framework does not fully capture whether the model's reasoning aligns with the expected solution paths. We agree that evaluating the trajectory of the reasoning process is crucial for understanding a model's problem-solving abilities.
>
>
> For now, we operate under the assumption that if the final answer is correct, the reasoning process is likely to be reasonable. Additionally, since we evaluate a sufficiently large number of questions, assessing mathematical reasoning ability based on the correctness of the final answer can effectively enhance the robustness of our evaluations.
>
>
> Regarding whether MathEval can evaluate the reasoning process, from a benchmark perspective, it is challenging to employ a model-based method for this task without introducing a significant number of misjudgments. Even an excellent PRM model with 95% accuracy could produce many errors due to the extensive reasoning steps involved in complex problems. Therefore, to maintain the fairness and reliability of the benchmark, we have temporarily decided not to include reasoning process evaluation.
>
> ## For weakness3
>
> Thank you for highlighting this concern. We agree that refining the classification of mathematical reasoning types can improve the depth and granularity of our evaluations. In future work, we plan to conduct more detailed classifications, such as categorizing problems based on algebraic manipulation, geometric reasoning, combinatorial logic, and other specific reasoning skills.

---

### Official Review · Reviewer_Nkro · 2024-11-08

**Soundness:** 2
**Presentation:** 2
**Contribution:** 2
**Rating:** 3
**Confidence:** 5

**Summary:**

MathEval is an extensive benchmark that includes various math scenarios, different types of prompts, and LLM-based evaluation. This solves the problem of incomprehensiveness, inadequate adaption, and inconsistency. MathEval incorporate 22 datasets in English and Chinese. MathEval also has dynamically updated content to prevent data contamination. In addition, it uses a robust evaluation framework, leveraging GPT-4 for automated comparison. It also uses its findings to fine-tune a DeepSeek-7B-based model and validates evaluation accuracy through human annotation. It also evaluates 52 models across three categories and proviees a comparative analysis.

**Strengths:**

Overall, much of what I mentioned in the summary is a strength of the paper, but here are some more specific strengths I have noticed:
- I very much appreciate how clear the introduction is. The the distinction between the three primary issues in evaluation the mathematical reasoning capabilities of LLMs is precise and clean.
- The experiments are very thorough, evaluation 52 models across 22 datasets.
- I appreciated the use the human annotation process to demonstrate precision.
- The two-stage evaluation approach combining GPT-4 and regex methods is great.
- I think the flexible prompt adaptation strategies for different model types is very useful for future research.
- Overall, this paper systematically addresses the challenge of mathematical answer comparison.

**Weaknesses:**

Here is a list of areas for improvement:
- I find it a bit difficult to read Figure 2 due to the acronyms. You could use icons for each category (e.g. flag icons for languages, school building icons for grade levels, etc.) or use a combination of icons and abbreviated text.
- Although Figure 3 is very informative, I find it a bit difficult to read the light green text on the gray background. I recommend updating the color scheme to dark text on light background or increasing the font size of the text within the boxes. You might also want to use a colorblind-friendly palette to ensure accessibility.
- For Figure 4, am I correct in saying that part c uses the results from parts a and b? If so, I recommend including arrows from parts and b to part c. This would help understand the flow of the components and the overall framework.
- Although it is great that MathEval has math problems from multiple levels of education, I am curious about why you don’t have more college level mathematics, at least at the undergraduate level. Ideally, this includes both academic math, such as real analysis, and competition math, such as the Putnam competition. I think this would provide more insights into how different models fare on different difficulty levels. Can you explain your rationale for focusing on K-12 levels and discuss the potential benefits and challenges of incorporating higher-level mathematics in future iterations of MathEval?
- I like how you mention details on what datasets are included in MathEval in the appendix. However, I think it would be beneficial to include a more detailed summary in the main paper. Otherwise, the reader may be left wondering what kind of data is in MathEval and how difficult it really is. I suggest including a brief summary table or paragraph in the main text that outlines key characteristics of the datasets (e.g. number of problems, difficulty levels, main mathematical concepts covered).
- You mention that you use calculation scheduling. However, I recommend including in the Appendix more details of the algorithm used and why it was chosen.
- I recommend including more middle school datasets (only Arith3K exists in MathEval at this point) for balance.
- Can you add some implementation details of the evaluation pipeline? This would aid reproducibility.
- I think training details for the DeepSeek-7B comparison model could be more detailed.
- Can you add more details on how the dataset is automatically updated?
- Can you include more details on how you generated the parts of MathEval that are labeled with “(ours)” in Table 2?

**Questions:**

- Would you have significantly different results if you used Claude rather than GPT-4 for the LLM-based evaluation? What are the limitations of using GPT-4 as an evaluator?
- Why did you choose Chinese as the second language to consider? Why not consider other languages? What unique advantage, if any, does analyzing Chinese language performance over other languages bring?
- What are the failure modes are error patterns that consistently appear because of dataset construction? - - What are some limitations?
- Why do chat models consistently outperform base models in these tasks?
- What types of mathematical reasoning are not currently captured by the benchmark?

---

> ### Author Response · Authors · 2024-11-15
>
> ## Figure-Related Issues
> **Q: Difficulty in Reading Figures Due to Acronyms, Color Scheme, and Flow of Components**
>
> Thank you for your insightful suggestions regarding Figures 2, 3, and 4. We agree that these improvements will significantly enhance the clarity and accessibility of our figures. Specifically:
>
> - For Figure 2, we added intuitive icons to represent different languages and grade levels. Specifically, we introduced language icons for various languages and used school building icons to distinguish between different grade segments. This visual enhancement improves the readability and intuitiveness of the chart.
> - Regarding Figure 3, we optimized the color scheme. We changed the previously hard-to-distinguish light green text to a prominent orange, while avoiding red-green combinations to ensure that individuals with color blindness can also read it easily.
> - For Figure 4, we added arrows pointing from parts a and b to part c, indicating that a portion of the training data for the compare-model was extracted and validated from the model's output as well as GPT-4’s answers. This improvement illustrates the logical relationships and information flow between the components, helping readers better understand the structure and workflow of the entire framework.
>
> These changes have been done in our revision version
>
>
> ## Dataset and Difficulty Settings
> **Q: Focus on K-12 Levels and Lack of Higher-Level Mathematics**
>
> Our current focus on K-12 education levels stems from their broader applicability to our user base and the availability of extensive datasets within this range. However, we recognize that incorporating higher-level mathematics, such as undergraduate topics (College Math) and competition math (e.g., PutnamBench), would offer deeper insights into the models' capabilities across varying difficulty levels. We are actively working towards including these more challenging problems in future iterations of MathEval.
> MathEval is actively maintained. Two months ago, we identified a gap in our benchmarks for competition-level problems, so we integrated OlympiadBench into our dataset collection. Since then, we have continued to expand our datasets and plan to include additional ones over time. Furthermore, we are preparing to encompass multimodal mathematical evaluations. To this end, we have already collected multimodal datasets, including MATHVISTA and MATHVERSE.
>
> We have added explatation in Appendix, Line 684-Line 688
>
> **Q: Inclusion of Dataset Summary in the Main Text**
>
> We will strive to include a dataset summary in the main text in subsequent versions of the paper.
>
> **Q: Adding More Middle School Datasets**
>
> We acknowledge that currently, Arith3K is the only middle school level dataset in MathEval. To achieve a balanced and diverse collection, we have recently collected the Zhongkao-2023 and Zhongkao-2024 datasets to compensate for the lack of middle school data. These datasets are derived from the Beijing High School Entrance Examination, which corresponds to the middle school level in China. Incorporating these datasets will enhance the diversity of MathEval and allow us to better evaluate models on mathematical abilities pertinent to middle school education.
>
> **Q: Details on Automatic Dataset Updates**
>
> From my point of view, this refers to our annual updates of the GAOKAO series datasets. As described in Appendix B.2 (line 680), these datasets are sourced from China's National College Entrance Examination (Gaokao). We have specialized educators who input the exam questions into our system shortly after the exam papers are released each year. By updating the datasets annually, we ensure that MathEval includes the most recent exam questions, which helps us evaluate models on fresh data and avoid potential data contamination from models that may have been trained on older testset.
>
> We have added a footnote to indicate that the dataset will updated in our Github Repo in Page 15.
>
> **Q: Generation of "Ours" Labelled Parts in Table 2**
>
> We have included the details of the generation of our datasets, such as SCQ-EN-5K, SCQ-CH-5K, GAOKAO, and Arith3K in Appendix B.1. To provide additional clarity, we have also shared the corresponding [GitHub](https://anonymous.4open.science/r/MathEval-505B/README.md) repository, where we offer more information about the dataset creation process.

---

> ### Author Response · Authors · 2024-11-15
>
> ## Evaluation and Algorithmic Details
>
> **Q: Details of Calculation Scheduling and details of the Evaluation Pipeline.**
>
> We will include the details of the calculation schedule in the appendix. This will cover dynamic dataset partitioning and automatic GPU allocation. For the evaluation pipeline, we have prepared our [anonymous github](https://anonymous.4open.science/r/MathEval-505B/README.md), which we believe will address these issues comprehensively.
>
> **Q: Training Details for the DeepSeek-7B Model**
>
> In the final version, we will provide more information about the training process for the DeepSeek-7B model. Due to anonymity concerns, we are unable to share our Hugging Face repository that contains all the details related to the training of the compare answer model. However, we utilized a straightforward Supervised Fine-Tuning (SFT) technique with standard language model loss for finetuning.
>
> ## Model and Language Choices
>
> **Q: Using Claude rather than GPT-4 for the LLM-based evaluation and limitations of using GPT-4.**
>
> We appreciate the reviewer's concern regarding our reliance on the GPT-4 model for answer comparison. Firstly, we would like to clarify our choice of GPT-4 over Claude-3.5. Our company has a collaborative agreement with the GPT-4 supplier, which grants us more efficient and cost-effective access to GPT-4, enabling extensive parallel processing capabilities. Our empirical evaluations suggest that the performance differences between GPT-4 and Claude-3.5 in answer verification tasks are minimal.
>
> We acknowledge that even the most advanced models, such as GPT-4 or Claude-3.5, do not yet match human performance comprehensively. Another limitation of using GPT-4 is its high cost. With over 70,000 questions across 52 models, if each question averages 1000 tokens, this amounts to approximately 36.4 billion tokens, resulting in significant expenses.
>
> **Q: Choice of Chinese as the Second Language to Consider.**
>
> We chose Chinese as the second language for several reasons. First, Chinese is extensively used worldwide, and there is a wealth of rich and representative math data available in Chinese. Second, as can be seen, a significant portion of the large language models we evaluate are trained by Chinese companies. In our view, analyzing Chinese does not offer a unique advantage over other languages. We selected Chinese based on the availability of data and the variety of models.

---

> ### Author Response · Authors · 2024-11-15
>
> ## Failure Modes and Other Limitations
>
> **Failure Modes or Error Patterns Due to Dataset Construction**
>
>
> First, we would like to clarify which stage of dataset construction is being referred to. If it pertains to prompt adaptation, we experimented with various prompt templates. We found that no Chain-of-Thought (no-CoT) prompts tend to introduce more problems compared to CoT prompts, especially in calculation problems that require step-by-step computations. For mathematical word problems like GSM8K, MATH, or datasets like OlympiadBench, the more reasoning steps required, the higher the likelihood of errors. This suggests that prompt design significantly impacts model performance, and carefully crafted CoT prompts are essential to mitigate error rates in complex reasoning tasks.
> If the concern is about the training data for our compare-answer model, we did not observe any significant error patterns.
>
> **Why Chat Models are Stronger**
>
>
> We provided a brief explanation in line 362 of the paper. Chat models generally undergo post-training based on base models, during which a substantial amount of data and methods specifically aimed at enhancing reasoning abilities are incorporated. For instance, models like DeepSeek-Math use techniques such as GRPO to improve their mathematical reasoning capabilities. Additionally, chat models are better at following instructions, which is a significant advantage over base models.
>
> **Mathematical Reasoning Not Captured by the Benchmark**
>
>
> Currently, our benchmark may not fully capture some advanced mathematical reasoning, such as problems involving diagrams or proof-based questions. We are actively preparing to include these two types of problems in future evaluations. While they may not be presented in this paper, we have conducted preliminary experiments.
> Regarding diagram-based problems, we have obtained initial results that will be part of a major update for MathEval. These results highlight significant challenges in multimodal mathematical reasoning, as models often struggle with interpreting and reasoning about visual information effectively.
> Here are the current performance metrics for various models on different benchmarks:
>
>
> | Model             | StatsChartMWP | MATHVISTA | MATHVERSE |
> |-------------------|---------------|-----------|-----------|
> | LLaVA-NeXT-34B    | 15.67         | 46.5      | 34.6      |
> | InternLM-XC2      | 17.13         | 57.6      | 27.4      |
> | Qwen-VL-PLUS      | 19.68         | 43.3      | 21.3      |
> | InternVL-1.2-Plus | 22.16         | 59.9      | -         |
> | GPT-4V            | 34.28         | 49.9      | 53.6      |
> | GPT-4o            | 55.62         | 63.8      | -         |

---

> > ### Comment · Reviewer_Nkro · 2024-12-02
> > **Response 1**
> >
> > I appreciate the authors’ response to my concerns. However, I find the response to be insufficiently robust for several reasons. The authors reiterate that their focus on K-12 levels is due to broader applicability and the availability of datasets. However, they fail to provide concrete evidence or data supporting the claim that K-12 levels are more broadly applicable to their user base. Are there user studies or usage statistics to back this focus? The integration of OlympiadBench is noted, but no specific details are given about what progress, if any, has been made toward incorporating undergraduate or Putnam-level mathematics. The authors reference their additions to the appendix (lines 684–688), but these edits merely restate the same general points made in the response without providing additional depth or insight. For instance, there is no mention of whether specific steps, such as pilot experiments with higher-level mathematics or user engagement studies, have been initiated. The mention of new datasets like MATHVISTA and MATHVERSE is intriguing, but it is not tied back to the primary concern: the lack of undergraduate and competition-level math. Are these datasets addressing that gap? If so, why were they not explicitly discussed?
> >
> >
> > While the appendix provides details, the main text should include a high-level summary to ensure the benchmark’s comprehensiveness is apparent. This could take the form of a table or brief paragraph outlining critical attributes, such as the number of problems, difficulty levels, languages, and key mathematical concepts covered. Without this addition, the paper’s impact is diminished, as it fails to adequately communicate the benchmark’s strengths to a broader audience.
> >
> > I appreciate the authors’ response regarding calculation scheduling. However, I find the reply to be insufficient. While the authors state they will include the details in the appendix, there is no evidence in the current manuscript of these details being present. I could not find a description of the algorithm for dynamic dataset partitioning or GPU allocation, either in the main text or in the appendix. The absence of these details is problematic, as calculation scheduling seems to be an important part of the benchmark’s implementation, particularly in large-scale computational tasks.
> >
> > I appreciate the authors’ acknowledgment of the imbalance in middle school datasets and their effort to address it by incorporating Zhongkao-2023 and Zhongkao-2024. However, these additions are not reflected in the current submission, despite the response being made before the deadline for paper edits. This makes it difficult to assess their impact on the diversity and balance of MathEval. Details such as the dataset characteristics, number of problems, and the specific mathematical concepts they cover are essential to evaluate their relevance and contribution.
> >
> > I appreciate the authors’ reference to a GitHub repository for implementation details of the evaluation pipeline. However, this response is insufficient. Providing a GitHub repository without adding relevant implementation details in the paper fails to address the concern about reproducibility. Reviewers and readers should not have to rely solely on external resources to understand the evaluation pipeline, particularly since repositories may be updated or removed over time, potentially compromising the reproducibility of results.
> >
> > I appreciate the authors’ response regarding the training details for the DeepSeek-7B comparison model. However, the explanation provided is vague and insufficient. While I understand the constraints of anonymity, the description of "straightforward Supervised Fine-Tuning (SFT) with standard language model loss" lacks the necessary depth to assess the training process. Key details, such as the size and characteristics of the training dataset, the number of training epochs, the optimizer used, hyperparameter values, and evaluation metrics during training, are missing. Without these details, the reproducibility and rigor of the training process cannot be evaluated.
> >
> > I appreciate the authors’ explanation regarding the annual updates to the GAOKAO datasets. However, the response does not adequately address my request for more details on the automatic dataset update process. The description provided focuses on manual input by educators, which contradicts the claim of an "automatic" update mechanism. There is no explanation of whether any automation exists, such as for data cleaning, formatting, or integration into MathEval.

---

> > ### Comment · Reviewer_Nkro · 2024-12-02
> > **Response 2**
> >
> > I appreciate the authors’ response regarding the use of GPT-4 for LLM-based evaluation. However, the reply is unsatisfactory for several reasons. The authors state that the performance differences between GPT-4 and Claude-3.5 Sonnet are minimal without providing any evidence or comparative analysis to support this claim. The authors mention a collaborative agreement with GPT-4's supplier, which has led to their reliance on GPT-4. While cost and access considerations are understandable, such agreements should not dictate the scientific validity of a benchmark. Evaluating alternative models, such as Claude, would provide a more robust and objective assessment and ensure that the choice of evaluator does not introduce bias into the results. The authors acknowledge that GPT-4 has limitations, including its cost and incomplete alignment with human performance. However, they do not discuss the potential impact of these limitations on the evaluation results or how they plan to mitigate them. The authors’ justification focuses on cost efficiency rather than exploring the scientific implications of using different LLMs as evaluators. This misses an opportunity to enhance the robustness and generalizability of MathEval by demonstrating consistency across multiple evaluators.
> >
> > I appreciate the authors’ attempt to address the question about failure modes and error patterns. However, the response is insufficient. While the authors mention prompt adaptation and the compare-answer model, they fail to directly address failure modes stemming from the dataset construction process itself. For example, are there biases, redundancies, or inconsistencies in the datasets that could systematically influence model performance or evaluation results? This was the primary focus of the question, and it remains unanswered. The authors reference datasets like GSM8K, MATH, and OlympiadBench, noting increased error rates with more reasoning steps. However, this is a generic observation rather than an analysis of specific dataset-driven error patterns. The claim that no significant error patterns were observed in the compare-answer model training data lacks sufficient evidence. Given the scale and complexity of the datasets, it is unlikely that no patterns or limitations emerged. What steps were taken to validate this claim? Was a thorough analysis of the error cases performed?
> >
> > Additionally, I have looked at the responses to other reviewers. I find a recurring issue across responses: they lack the depth and specificity necessary to address the concerns raised. For instance, when reviewer 7NzK pointed out the rough classification of mathematical reasoning types, the authors merely acknowledged the concern and deferred its resolution to future work. Acknowledging issues without taking substantive action or providing detailed plans risks diminishing the credibility of the paper.
> >
> > Overall, I find the response to be insufficiently robust for several reasons noted previously. Given the consistent lack of depth and rigor in the responses and subsequent versions of the paper, I have decided to lower my score from a 6 to a 3. While the paper addresses an important area, it does not adequately address essential feedback issues. Its inability to demonstrate scientific rigor significantly undermines its quality and potential impact.

---

### Author Response · Authors · 2024-12-01
**Request for Feedback on Responses**

Dear Reviewers,

We would like to express our gratitude for your meticulous review of our paper. We have provided responses to the comments and concerns you raised.

As the discussion period comes to a close, we would appreciate your feedback on whether our responses have addressed the issues raised and whether they may lead to an improvement in the scores. We also welcome any new questions or continued discussions.

Thank you again for your hard work and support.

Best regards,

The authors of Paper 6578

---

### Note · Authors · 2025-01-23

I have read and agree with the venue's withdrawal policy on behalf of myself and my co-authors.